# Connecting spatial and temporal scales of tropical precipitation in observations and the MetUM-GA6

Gill M. Martin[1], Nicholas P. Klingaman[2] and Aurel F. Moise[3]

[1]Met Office, Exeter, United Kingdom
[2]National Centre for Atmospheric Science-Climate and Department of Meteorology, University of Reading, United Kingdom
[3]Bureau of Meteorology, Melbourne, Australia

*Correspondence to*: Gill Martin (gill.martin@metoffice.gov.uk)

**Copyright statement**

**Abstract.** This study analyses tropical rainfall variability, on a range of temporal and spatial scales, in a set of parallel Met Office Unified Model (MetUM) simulations at a range of horizontal resolutions, compared with two satellite-derived rainfall datasets. We focus on the shorter scales i.e. from the native grid and time-step of the model through sub-daily to seasonal, since previous studies have paid relatively little attention to sub-daily rainfall variability and how this feeds through to longer scales. We find that the behaviour of the deep convection parametrization in this model on the native grid and time-step is largely independent of the grid-box size and time-step length over which it operates. There is also little difference in the rainfall variability on larger/longer spatial/temporal scales. Tropical convection in the model on the native grid/time-step is spatially and temporally intermittent, producing very large rainfall amounts interspersed with grid-boxes/time-steps of little or no rain. In contrast, switching off the deep convection parametrization, albeit at an unrealistic resolution for resolving tropical convection, results in very persistent (for limited periods), but very sporadic, rainfall. In both cases, spatial and temporal averaging smoothes out this intermittency. On the ~100 km scale, for oceanic regions, the spectra of 3-hourly and daily mean rainfall in the configurations with parametrized convection agree fairly well with those from satellite-derived rainfall estimates, while at ~10 day timescales the averages are overestimated, indicating a lack of intra-seasonal variability. Over tropical land the results are more varied, but the model often underestimates the daily mean rainfall (partly as a result of a poor diurnal cycle) but still lacks variability on intra-seasonal timescales. Ultimately, such work will shed light on how uncertainties in modelling the small/short scale processes relate to uncertainty in climate change projections of rainfall distribution and variability, with a view to reducing such uncertainty through improved modelling of the small/short scale processes.

## 1 Introduction

The realism of rainfall in a climate model is a key indicator of its skill in representing the underlying physical processes, and hence in increasing our confidence for projecting future changes in rainfall. In particular, the spatial and temporal structure of rainfall variability is arguably a more important indicator of this skill than the absolute rainfall amount or the aggregated mean state, which is typically used to assess models. When and where rainfall will occur, and with what intensity and duration, is essential information, particularly in vulnerable tropical regions where the livelihoods of millions rely on seasonal, rainfall-driven agriculture and where infrastructure is often lacking, even when extremes associated with rainfall variability (droughts, floods) are relatively commonplace. Although there have been some improvements in some aspects of the representation of precipitation between the Third and Fifth phases of the Climate Model Intercomparison Project (CMIP3; Meehl et al. (2007) and CMIP5; Taylor et al. (2012), respectively), as described, for example, in Koutroulis et al. (2015), uncertainties in hydrological predictions from the current generation of models still pose a serious challenge to the reliability of projections across temporal and spatial scales (Trenberth, 2011).

Previous studies have highlighted that climate model biases on multi-year and global scales develop within a few days of the start of the simulation (e.g., Martin et al., 2010) and are closely related to deficiencies in the simulation of processes on much shorter and smaller scales (e.g., Stephens et al., 2010). Such mean state biases in rainfall can be associated with other biases such as in sea surface temperatures (e.g., Levine and Turner, 2012) and can contribute to uncertainty in projections of future tropical rainfall (e.g., Kent et al., 2015). Deep convection parametrizations in these models often produce very intermittent rainfall at the level of the model's time-step and grid-scale, and also produce a poor representation of the processes and timing associated with the diurnal cycle of convection over land (e.g., Stratton and Stirling, 2012). Such deficiencies can have a significant impact on the regional-scale circulation and water cycle (e.g., Birch et al., 2014). Studies such as Kendon et al. (2014) illustrate that representing rainfall characteristics on short and small scales may be paramount in order to eliminate these biases and thereby provide confidence in projections of the spatial and temporal characteristics of heavy rainfall in a future climate.

The sheer volume of data required for analyses of rainfall variability at sub-daily timescales and kilometre-scale resolutions can deter model developers. Analysis of sub-daily rainfall variability is therefore relatively limited in the scientific literature. However, the importance of studying changes in the location, type, amount, frequency, intensity, and duration of precipitation, and especially to changes in extremes, has been highlighted by several authors in recent years (e.g., Trenberth, 2011; Tripathi and Dominguez, 2013; Cortez-Hernandez et al., 2015). Studies of sub-daily rainfall and extremes in observations (e.g., Westra et al., 2014) and in models (e.g., Rosa and Collins, 2013) are becoming more common in the literature and highlight discrepancies between models and observations and sensitivities to model resolution and physical parametrizations.

Klingaman et al. (2016) showed how these large data volumes can be condensed to a manageable set of diagnostics (Analysing Scales of Precipitation Version 1.0, ASoP1) with which we can both increase understanding of observed rainfall

variability and compare model behaviour on a range of time and space scales. The ASoP diagnostics include correlations with distance and time, as well as one-dimensional and two-dimensional spectra of rainfall amounts, and can be applied to data on any time or space scale, although the diagnostics are designed for the range from time-step/grid-scale up to sub-seasonal/meso-$\alpha$ scale (~90 days/~500 km). Klingaman et al. (2016) applied ASoP1 to Indo-Pacific Warm Pool precipitation data from ten models used in the "Vertical structure and physical processes of the Madden-Julian oscillation" model-evaluation project (Xavier et al. 2015). The authors found large inter-model variations in the degree of spatial and temporal intermittency in time-step precipitation, but that the models' scales of precipitation were highly similar when the precipitation data were averaged to the 3 h, 600 km scale.

Motivated by those results, in the present study we use the ASoP1 methods to examine how the spatial and temporal intermittency of tropical precipitation in the Met Office global general circulation model (GCM) varies with horizontal resolution - and, by extension, time-step length - and the treatment of deep convection. In Klingaman et al. (2016), the Met Office Unified Model (MetUM) displayed particularly high spatial and temporal intermittency in time-step and grid-scale precipitation. Here, we analyse sub-daily precipitation intermittency in the simulations across a range of horizontal resolutions with parametrized convection, as well as in a simulation with an explicit representation of mid-level and deep convection. In all cases, we examine how sub-daily precipitation intermittency may influence rainfall characteristics at longer timescales (up to ~20 days), in order to demonstrate how the ASoP1 diagnostics can be used routinely as part of model parametrization development.

The paper is arranged as follows: in section 2 we provide details of the model and observation datasets used in this study; in section 3 we analyse the temporal and spatial coherence of the tropical rainfall on sub-daily timescales; in section 4 we examine the spectral distributions of rainfall amounts at a range of time and space scales up to ~20 days; our discussion and conclusions are presented in section 5.

## 2 Datasets used in this study

### 2.1 Model description

We use MetUM Global Atmosphere 6.0 (MetUM-GA6; Walters et al., 2016; Williams et al., 2015), which is an updated version of the MetUM-GA3 (Walters et al., 2011) configuration analysed by Klingaman et al. (2016), with a different dynamical core (ENDGAME; Wood et al., 2013), a new orographic gravity wave drag representation (Vosper et al., 2009), and several changes to the convective parametrization (see Walters et al. (2011) and Walters et al. (2016) for details). MetUM-GA6 includes a 25% increase to the rates of mixing entrainment and detrainment for diagnosed deep convection relative to MetUM-GA3, implemented to improve the representation of tropical sub-seasonal variability following Klingaman and Woolnough (2014). MetUM-GA6 atmosphere-only simulations are forced with daily observed sea surface temperature (SST) and sea-ice forcings from the OSTIA product (Donlon et al., 2012), bilinearly interpolated from the OSTIA 1/20° resolution to the MetUM horizontal grids.

The MetUM-GA6 naming conventions and parameter settings for the different resolutions used in the current study match those described by Johnson et al. (2015) for MetUM-GA3. As discussed by Johnson et al. (2015), very few parameters in the MetUM are changed with resolution, but there are a few that must be changed to ensure numerical stability. In the MetUM-GA6 simulations analysed in our study, most of the parameter settings for the different resolutions match those shown in Johnson et al. (2015; their Table 2). Note that the inclusion of the ENDGAME dynamical core improved model stability, negating the need for targeted diffusion of moisture. An additional resolution, termed "N1024", with a 0.18° x 0.12° grid, is also included. The settings at N1024 resolution are also kept the same, except for the dynamical core's alternating-direction implicit (ADI) pseudo time-step, which is related to the efficiency of the implicit solver at high latitudes. This is reduced to $7 \times 10^{-5}$ in the N1024 simulations.

Table 1 contains details of the MetUM-GA6 simulations and the domains over which we analyse the precipitation data. Time-step rainfall data for an extended tropical region (40°S-40°N) were archived for only one June-September season (JJAS) due to their computational and storage costs. The year of output depended on when the time-step diagnostics were enabled manually; for the N512 simulation this was originally 1985. However, due to a technical error in the original diagnostic output, the simulation had to be repeated, using the same configuration and with time-step diagnostics enabled, for June-September 2007, due to the availability of a 1st June restart file for that year. Daily data were available for at least 8 years (often 27 years), from 1982 onwards, in all but the N1024 simulations, which were run for only 4 years due to computational cost. Due to the relatively small amount of daily data available for the N1024 simulations, only the time-step data for these configurations are included in this study. Despite time-step data being available for differing years between the runs, we consider that the sample is sufficiently large for the results to be robust. Comparison of other model runs (not shown) where more than one season of time-step data was available have also shown that the results have little sensitivity to the year used. In the analysis of spatial and temporal intermittency, for most of the simulations we analyse a tropical domain covering the Equatorial Indian Ocean, Maritime Continent and the far western Pacific Ocean (10°S–10°N, 60–160°E), hereafter the "EQ" domain. For the spectral analyses we use a larger domain covering 20°S–40°N, 20°W–160°E. For the highest-resolution (N1024) simulations, we use data only over the two limited domains that were available to us (due to storage and computational limits), one in the western Pacific Ocean (0°–20°N, 130–160°E; hereafter the "WP" domain), and the other over West Africa (8°–17°N, 0°–10°E, hereafter the "WA" domain). We use data from the next-finest resolution (N512) to demonstrate that there are limited differences in sub-daily precipitation characteristics over ocean between the EQ and WP domains. The domains used are illustrated in Fig. 1.

As an additional test of the ability of the ASoP1 diagnostics to identify different behaviour in rainfall variability, we also analyse a N1024 simulation where parametrized deep convection is switched off (N1024e). Although this simulation has a horizontal resolution at which explicit convection is unlikely to be realistic, it is worth exploiting the opportunity afforded by this pair of simulations to compare the rainfall variability with and without parametrized deep convection, but at the same horizontal resolution and model time-step.

## 2.2 Satellite-based rainfall analyses

We compare the models' precipitation data with two sets of satellite-derived analyses: those from the Tropical Rainfall Measuring Mission 3B42 product, version 7 (TRMM; Kummerow et al. 1998a; Huffman et al. 2007a) and those from the CPC MORPHing technique version 1.0 (CMORPH; Joyce et al. 2004). Both products are derived from a combination of infrared and microwave sounders and calibrated against gauge data. TRMM and CMORPH are available at 3-hourly and daily time resolution and a maximum horizontal resolution of 0.25° x 0.25° resolution. We analyse daily averages of these products across a common period of 2001-2012, while JJAS from the year 2005 is used for analysis of the raw 3-hourly data, for comparison against the single JJAS of 3-hourly data from each model configuration. Comparisons of the results for 3-hourly data between this single JJAS season and all JJAS seasons from each dataset show only small differences (not shown), confirming that the use of a single season is justified.

## 3 Sub-daily spatial and temporal intermittency

### 3.1 Behaviour on the native grid and time-step

Two-dimensional (2D) probability distribution functions (PDF) of binned grid-box precipitation in time interval $t$ against precipitation in the next interval $t + 1$ are used to diagnose the behaviour of satellite-derived and simulated precipitation between consecutive temporal intervals at a fixed horizontal point (see Klingaman et al. (2016) for details of the methodology). When applied to time-step data on the native grid from MetUM-GA6 simulations with parametrized convection (Fig. 2a-e), these PDFs show higher probabilities along the axes and lower probabilities on the central diagonal. This demonstrates that, with parametrized convection, MetUM-GA6 produces substantial temporal intermittency in time-step, grid-box precipitation as heavy precipitation on one time-step is followed by light or no precipitation on the next time-step, and vice versa. There is very little variation in this behaviour with horizontal resolution and time-step length: over the EQ domain, N512 rainfall similarly intermittent to N96 rainfall, despite a ~25-fold reduction in grid-box area; over the WP domain, N1024p rainfall is similarly intermittent to N512 rainfall, despite a four-fold reduction in grid-box area. Comparing N512-EQ and N512-WP demonstrates that temporal intermittency in rainfall is similar in these regions, suggesting that N1024p-WP can be compared with the coarser-resolution models over the EQ domain.

Perhaps most striking is the consistency of the time-step rain-rate PDFs (dashed line) among the parametrized-convection configurations, regardless of the horizontal resolution. Resolution hardly alters the PDF of time-step rainfall, when converted to daily rates, which indicates that the convective parametrization is not strongly affected by changes in grid-box area or the associated changes in the strength of the dynamical forcing. We hypothesize that this is due to an "all-or-nothing" behaviour in the MetUM convective parametrization: when deep convection is triggered, the parametrization often produces the maximum possible rain rate, even for relatively weak forcing. This is consistent with the lack of moderate time-step rain rates (9-30 mm day$^{-1}$) at all resolutions with parametrized convection. We note that the rain-rate PDFs would

differ with resolution if we expressed the rain rates as time-step$^{-1}$ values, but this would not provide a clean comparison between the simulations.

The intermittent behaviour in the tropical deep convective rainfall is caused by the choice of closure at GA6, in which the mass flux amplitude is set to depend on the CAPE detected in the grid-box, rather than the rate of atmospheric destabilisation (A. Stirling, personal communication). The resultant heating applied produces an inversion at the top of the boundary layer on the next time-step that the diagnosis deems too strong to allow convection to initiate. It remains in this state until the inversion has been eroded by a combination of heating in the boundary layer and large-scale ascent. Examination of timer-series of tropical rainfall from the start of each simulation (not shown) indicates that this behaviour occurs immediately at the start of the simulation with very little spin-up (less than 1 day), regardless of grid size or time-step length.

Switching from a parametrized to an explicit treatment of deep convection at N1024 resolution transforms MetUM-GA6 from producing highly intermittent precipitation (Fig. 2e) to highly persistent precipitation (Fig. 2f). In the 2D PDF of time-step, grid-box precipitation, N1024e produces high values on the diagonal and low values on the axes, reminiscent of the most persistent models analysed by Klingaman et al. (2016). However, the highly bimodal 1D rain-rate PDF (dashed line in Fig. 2f) shows that N1024e exhibits even stronger "all-or-nothing" behaviour than N1024p. On average, only 2% of time-steps precipitate at rates $\geq 2$ mm day$^{-1}$, but most of those have rates $\geq 180$ mm day$^{-1}$. This is almost certainly due to the extremely strong forcing required to lift a ~20 km x 14 km grid-box, and confirms that N1024e is a very coarse resolution at which to use an explicit representation of deep convection. Future work will investigate how this behaviour changes as the resolution is increased in convection-permitting simulations.

ASoP1 measures precipitation coherence as a function of the native time-step and grid by dividing the analysis region into 7x7 sub-regions, computing lag correlations of each grid-box in the region against the central grid-box, then compositing these correlations across all sub-regions (see Klingaman et al. 2016 for details). For ease of display, the spatial correlations are binned by the distance away from the central grid-box, in units of the longitudinal grid spacing at the equator ($\Delta x$). Table 2 gives the number of 7x7 sub-regions in each model and region. In MetUM-GA6, all parametrized-convection resolutions show similarly low coherence in time-step, grid-box precipitation to that found by Klingaman et al. (2016) for MetUM-GA3, with a lag-1 minimum in the auto-correlation at the central grid-box that indicates a preference for "on-off" convection (Fig. 3a-e). The fact that the correlations between surrounding grid-boxes and the central grid-box are essentially constant at all lags shows that convection at the surrounding grid-boxes evolves independently of the central grid-box, confirming a lack of spatial organisation. Switching to an explicit representation of convection in N1024e produces temporally consistent precipitation at the central grid-box (Fig. 3f), but does not improve the low spatial coherence of rainfall, which is reduced further compared with N1024p. This is because convective heating associated with explicit convection sets up significant ascent in the convecting column, which continues the destabilisation of the column, while adjacent columns experience descent and so convection is suppressed.

## 3.2 Effects of temporal averaging

To examine whether the characteristics of grid-box, time-step precipitation discussed in section 3.1 persist at longer timescales, we apply the 2D histogram diagnostic from Klingaman et al. (2016) to 3 h averaged time-step precipitation data (Fig. 4). Such temporal averaging reduces precipitation intermittency at all resolutions with parametrized convection, producing higher probabilities along the central diagonal and lower probabilities along the axes relative to Fig. 2. This implies that, when averaged over three hours, the convection scheme starts to display sensitivity to the large-scale forcing, as the strength thereof determines the frequency with which the convection scheme can be activated. In contrast, such averaging leads to much greater intermittency for the N1024e configuration (Fig. 4i). The temporal persistence seen in the N1024e time-step data (Fig. 2f) does not carry across to the 3-hourly scale, likely because the decorrelation time of grid-scale precipitation in the explicit-convection configuration is much longer than a time-step (5 min) but shorter than 6 h (i.e., two consecutive 3 h periods, as considered in the 2D histograms). This suggests that the grid-box precipitation features in N1024e, and the associated grid-scale forcing, often have lifetimes of 3 hours or fewer.

N1024e-WP also has consecutive 3 h steps with very high (>180 mm day$^{-1}$) rainfall, which occurs about 35% of the time that there is rainfall is in this bin (i.e., 35% of the time that there is >180 mm day$^{-1}$ in one 3 hour window, there is also >180 mm day$^{-1}$ in the next 3 h window). All configurations with parametrized convection produce 3 h rain rates that are too persistent relative to CMORPH and TRMM, whether the analyses are considered on their native grids (Fig. 4a-c) or averaged to the same grids as the model configurations (shown for CMORPH only, Fig. 4j-l). Comparing CMORPH across resolutions shows an increase in precipitation intermittency at finer grid-scales, which MetUM-GA6 also shows, but to a more limited extent. TRMM rainfall is somewhat more intermittent than CMORPH, but the results from all model configurations are outside the range of those from the satellite-derived analyses.

## 3.3 Effects of spatial averaging

To test the effects of spatial averaging on the characteristics of time-step precipitation, we average the data from each model to a horizontal resolution of 3.75° ✕ 2.5°, which is exactly 2 ✕ 2 N96 grid-boxes and equivalent to the MetUM N48 resolution. We refer to this resolution as "N48". We use N48 to ensure that all models are subject to some degree of spatial averaging, following Klingaman et al. (2016). Table 2 shows the number of native-resolution grid-boxes in each 3.75° ✕ 2.5° region for each model. Spatial averaging reduces temporal intermittency in precipitation at all resolutions, whether with parametrized or explicit convection (compare Fig. 5 with Fig. 2). All configurations produce higher probabilities on the central diagonal and lower probabilities on the horizontal and vertical axes. The reductions in intermittency are greatest for the finest-resolution configurations, with N512 (Fig. 5c) showing much more persistent precipitation than N96 (Fig. 5a) over the EQ region. This is due to the much greater number of N512 grid-boxes (113 boxes) averaged together to create each N48 grid-box, compared with N96 (4 boxes). Applying 2 ✕ 2 spatial averaging to N512 yielded a highly similar 2D PDF as for the N96 simulation averaged to N48 (not shown). Even at N48 resolution, when 450 boxes are averaged together, the

precipitation from the N1024e configuration (Fig. 5f) remains more persistent than that from the N1024p configuration (Fig. 5e), with far fewer precipitating grid-boxes.

## 3.4 Effects of temporal and spatial averaging

In a similar manner to the results of Klingaman et al. (2016), we find that applying temporal and spatial averaging to 3 h and ~400 km resolution, respectively, leads to similar 2D PDFs for all resolutions of this MetUM configuration that have parametrized convection, and that these are all too persistent relative to TRMM and CMORPH at the same resolutions (Fig. 6). The rain-rate PDFs are also remarkably similar between the resolutions, except for slightly more frequent heavy rainfall (and fewer near-zero values) at finer resolutions. In contrast, following temporal and spatial averaging, the configuration with explicit convection strongly resembles CMORPH and TRMM in temporal persistence and rain-rate PDF, except for having more near-zero values and fewer heavy-rain values. This is discussed further in section 4.3.

## 3.5 Correlations with physical distance and time

To summarize the spatial and temporal coherence of grid-box, time-step precipitation in the model configurations, as well as the effects of spatial and temporal averaging on that coherence, we present correlations of precipitation as functions of physical distance (in km) and time (in minutes). These diagnostics allow the model results to be compared more easily than in Fig. 6, because they show correlations as functions both of the number of model grid-boxes or time-steps and of physical distance and time.

To compute correlations as a function of physical distance, we divide the EQ and WP domains into equal-area regions, then correlate the rainfall in each grid-box in each region against the central grid-box in the region; correlation values are binned by the distance from the central grid-box, with a bin width equal to $\Delta x$ (see Klingaman et al. (2016) for details). In the EQ region, we use 1700 $\times$ 1700 km regions so that the region is at least $4\Delta x$ wide at the coarsest resolution considered (N48). In the WP region, we use 600 $\times$ 600 km regions for native-resolution data, due to the limited size of the region, but 1700 $\times$ 1700 km regions for N48 data for the reasons discussed above. Table 2 gives the number and dimensions of the equal-area regions for each MetUM-GA6 resolution in each region, including the N48 averaged data, as well for TRMM and CMORPH.

Correlations with distance show that all configurations with parametrized convection produce similar spatial scales of time-step precipitation, regardless of resolution, whereas the N1024e configuration produces very fine-scale features (Fig. 7a). In combination with the 2D histograms of time-step precipitation in Fig. 2, Figure 7a emphasises that refining horizontal resolution does not fundamentally alter the nature of parametrized convection in the MetUM. Averaging time-step precipitation in either space (to N48, Fig. 7b) or time (to 3 h means, Fig. 7c) increases the spatial coherence of precipitation, particularly for the finer-resolution models in which more grid-boxes or time-steps are averaged together. For 3 h mean rain rates, the MetUM-GA6 configurations with parametrized convection produce broader precipitation features than either

TRMM or CMORPH in both the EQ and WP regions, whereas the configuration with explicit convection shows much smaller-scale features than the satellite analyses.

Correlations with time show that all configurations with parametrized convection show a strong lag-1 decrease in the auto-correlation of time-step precipitation (Fig. 7e), which persists even when spatial averaging is applied (Fig. 7f), although it reduces in magnitude as more grid-boxes are averaged together in the finer-resolution models. This further demonstrates the intermittent nature of parametrized convection in MetUM-GA6, which is insensitive to horizontal resolution. The N1024e configuration produces a smooth auto-correlation function for time-step data, which asymptotes to the same value as the N1024p configuration within the 3 h window considered. While the two configurations have similar temporal coherence of precipitation at a 3 h lag, the N1024p configuration achieves this by averaging intermittent time-step convection while the N1024e configuration achieves this by averaging persistent convective events of various lifetimes ranging from a few time-steps to several hours.

## 3.6 Summary metrics

Table 3 presents summary metrics from ASoP1 of the spatial and temporal coherence in precipitation using time-step and 3 h data on the native horizontal grids and averaged to the N48 grid. These metrics are computed from the coherence of upper-quartile and lower-quartile precipitation in space and time; higher positive values indicate greater coherence. The metrics reflect the findings above: all models with parametrized convection show temporal intermittency in time-step data on the native grid, regardless of model resolution; spatial coherence is also low. After averaging to 3 h scales, on the models' native grids, there is a large increase in temporal and spatial coherence of tropical rainfall in all model simulations with parametrized convection. In this case, there is a noticeable increase in spatial coherence as model resolution increases (reflecting the decreasing grid-size, since the metrics are computed on the native grid), while there remains no systematic change in temporal coherence with resolution. The spatial coherence at this timescale is similar to that in the satellite-derived datasets, while the temporal coherence is noticeably greater.

Both the satellite-derived rainfall datasets and the model simulations show a reduction in spatial coherence of tropical rainfall at the 3 h timescale following averaging to the coarser N48 resolution, while the temporal coherence increases. The reduction in spatial coherence at N48 relative to the native grids occurs because the metrics are computed based on a fixed distance in grid-points (of the input data), rather than a fixed physical distance; a coarser grid would be expected to have less coherent precipitation, due to the greater physical distance between grid-points. On the N48 scale there is a more systematic increase in temporal coherence at finer model resolutions, particularly for the time-step data (partly, at least, due to the decreasing time-step with increasing model resolution). Overall, the values show that, following spatial and temporal averaging, the simulations with parametrized convection show only slightly larger spatial and temporal coherence than the satellite-derived rainfall, and little systematic change in either temporal or spatial coherence with resolution.

The metrics also illustrate the contrasting behaviour of the N1024e configuration. This shows high temporal and spatial coherence at the time-step, grid-scale level, consistent with Fig. 3f which suggested persistent rainfall isolated to one or two

grid boxes in this configuration. Spatial coherence remains similar upon temporal averaging but decreases sharply upon spatial averaging, due to the isolated nature of explicit convection at this grid-size. The temporal coherence of the time-step data remains high following spatial averaging but decreases upon temporal averaging (suggesting the events rarely last for longer than 3 h and are followed by prolonged dry intervals).

Ultimately, following averaging to the N48 grid and 3 h scale, the temporal and spatial coherence of tropical rainfall from all the model simulations is similar to that from the satellite-derived datasets.

## 4 Spectral characteristics

To examine the distribution of precipitation intensity on a range of spatial and temporal scales, and its sensitivity to temporal and spatial averaging, we compute the contributions of discrete bins of precipitation intensity to the total precipitation at a

grid-box. The result is a spectrum that shows the relative importance of precipitation events in a given intensity bin to the total precipitation. As in Klingaman et al. (2016), we use 100 bins of varying width, defined by the following equation and sampling rainfall intensities in the range 0.005 to 2360 mm day$^{-1}$:

$$b_i = e^{\left\{\ln(0.005)+\left[i.\frac{(\ln(120)-\ln(0.005))^2}{59}\right]^{\frac{1}{2}}\right\}}, \tag{1}$$

where $i$ is the number of the bin and ranges from 1 to 100, and $\ln(x)$ is the natural logarithm of $x$. A further lower bin edge

is added at 0.0 to ensure that a histogram of counts computed using these bins sums to the number of valid data points in the sample. By calculating these contributions at many grid-boxes in a region, we produce maps of the contributions of various precipitation intensity bins to the total precipitation at each grid-box (e.g., Fig. 8). Regional averages of the spectra can also be produced for direct comparison between datasets, although it should be noted that this process introduces a spatial averaging of the spectra themselves and hence is best done for relatively small regions only.

Analysis of the spectral characteristics of these runs provides further evidence of the points made in section 3, and allows us to investigate the influence on rainfall amounts at longer timescales. They also illustrate the effects of temporal and/or spatial averaging of time-step/grid-box data, which can indicate temporal and spatial intermittency. We analyse the spectral characteristics on each given timescale at each grid-box of the dataset (at whichever resolution is being analysed). We then use spectra averaged over particular regions to illustrate the characteristics of this model configuration. The regions were

chosen based on typical climatological bias regions illustrated in Walters et al. (2016): wet bias regions of the equatorial Indian Ocean, South China and the western Pacific, and the West Africa dry bias region.

### 4.1 Influence of resolution

We first compare the rainfall spectra at the native grid and time-step among model resolutions. Noting that the time-step length is shorter at the higher horizontal resolutions (see Table 1), the broad similarity, particularly over the ocean, between

the spectral maps at native resolution, even as the horizontal resolution is increased 5-fold, is remarkable (Fig. 8) and suggests that the convection parametrization behaviour in the tropics is not very scale-aware (except perhaps for larger rainfall events). Closer examination of the spectra in different regions (Figs 9a and 9b, solid lines) reveals that the higher resolution simulations do, as expected, produce more frequent, higher intensity time-step events on the native grid, particularly for the land region of West Africa (WA domain; Fig. 9b). However, when the higher resolutions are all averaged to N48 (Figs 9a and 9b, dashed lines), the rainfall spectra are shifted to smaller intensities in all cases and the differences in the tail of the distribution are no longer apparent, suggesting that those events were spatially isolated. We also note that the effects of spatial averaging are larger for the higher model resolutions in which more grid boxes are included in the average. This illustrates the spatial intermittency that was highlighted in Section 3. However, the largest impact of spatial averaging is seen for the N1024e configuration. This is discussed further in Section 4.3.

When the precipitation data are all averaged to the N48 grid and 3 h time scale (in a similar manner to Section 3.4), Figure 9c–f shows that the models all tend to underestimate the 3-hourly rainfall amounts compared with TRMM and CMORPH, and that increasing the horizontal resolution does not improve the comparison on this timescale for tropical rainfall over the ocean. Indeed, we see little evidence that increasing the horizontal resolution has any overall effect on the rainfall distribution over ocean on these scales in the simulations with parametrized convection. Over the West African land region the higher resolution configurations do show an increase in the fractional contribution from higher 3 h totals and a corresponding decrease in the lower amounts, but this is not apparent over South China. We note that the largest change in the spectral characteristics is seen when the convection parametrization is switched off. This is discussed further in Section 4.3.

## 4.2 Looking across timescales

We next examine how the spread of rainfall amounts changes as the data are averaged to successively longer timescales. We continue to examine the datasets once averaged to the N48 grid in order both to compare spectra at the same effective resolution and to ensure that at least some horizontal averaging has been done on all datasets. Data from the N96, N216 and N512 simulations, for June-September inclusive, are averaged to 3-hourly, daily, 10 day and 20 day timescales, over the years shown in Table 1. CMORPH data are also used for comparison, using the years mentioned in Section 2. Two oceanic regions and two land regions are selected in order to highlight the main findings.

The movement of the spectra towards smaller values when averaged to successively longer timescales indicates that there is variability at the longer timescale (such that including drier periods in the average decreases the longer timescale mean). For all of the regions shown in Fig. 10, the progressive shift of the spectra from CMORPH towards smaller values as the timescale is increased from 3 h to 20 days is not matched by the model results, which tend to show less movement and therefore lack variability on the longer timescales. This is particularly noticeable for tropical rainfall over the oceans: for the two regions shown in Fig. 10a and 10c the N96 configuration underestimates the rainfall totals at shorter timescales but overestimates them at longer timescales. The spectra of daily mean values agree reasonably well with those from CMORPH,

but the model lacks variations on timescales of ~10 days so that the spectra for the 10 day and 20 day means peak at larger values in the model than in CMORPH. This is confirmed in Figure 11a and 11c where auto-correlations of daily rainfall with increasing time-lag for these regions are consistently higher in the model than in the satellite rainfall datasets. For South China, the daily rainfall spectrum tends towards large values, suggesting a lack of sub-daily variability, while the day to day variability is in reasonable agreement with the observations (Fig. 11d). All three of these regions exhibit positive rainfall biases in the MetUM-GA6 configuration's climatology (Walters et al., 2016). In contrast, the West African region (Fig. 10b) shows spectra on all timescales that are displaced to smaller rainfall totals than CMORPH, consistent with a climatological dry bias. For this region, there is disagreement on the day to day variability between the two satellite-based rainfall estimates, with TRMM suggesting more persistence (higher autocorrelations) than either CMORPH or the model. These differing characteristics between the two measures of actual rainfall amounts are likely related to their different satellite data sources and the algorithms used to combine those sources. It is known that both datasets tend to underestimate smaller daily rainfall totals and can overestimate larger ones (e.g., Tian et al., 2010), but details on the day to day variability of rainfall in these two datasets are lacking in the literature. Thus, we cannot make a definitive statement about the validity of the characteristics of daily rainfall variations in the models compared with satellite-based estimates over the West African region.

Figures 11 and 12 show the same comparison but for the N512 configuration. As indicated in Section 4.1, this 5-fold increase in horizontal resolution has little consistent impact on these characteristics of tropical rainfall variability. Comparison across these timescales of spectra derived from these two configurations at their native resolutions, compared with CMORPH data averaged to each of those model grids (not shown), indicates a similar lack of consistent improvement at the higher resolution.

## 4.3 Explicit vs parametrized convection

The results presented above suggest that, generally, MetUM-GA6 configurations with the deep convection parametrization switched on have similar spectral characteristics across timescales despite differing grid sizes and time-steps. Only once the deep convection scheme is switched off do the characteristics change markedly. The N1024e simulation with explicit convection produces extremely high intensity time-step events which persist for up to 3 hours (Fig. 3f). Thus, in contrast with the N1024p simulation, there is virtually no difference between the spectra from the N1024e time-step data and 3 h averages on the native grid (light and dark green curves on Fig.s 13a and 13c), showing limited effects of temporal averaging. However, as illustrated in Fig. 3f and Fig. 7a, the high-intensity time-step events are isolated to only one or two grid-boxes any given time, so spatial averaging has a large impact (compare the pairs of green and purple lines in Fig.s 13a and 13c). Further, once spatial averaging has been carried out, the subsequent effects of temporal averaging between time-step and 3 h scales are negligible (the light purple curves in Fig.s 13a and 13c are almost hidden by the dark purple curves). In contrast, for N1024p (see Fig.s 13b, 13d) there are effects from both temporal and spatial averaging of the time-step data

because they are both temporally and spatially intermittent. However, once again, following spatial averaging of the time-step data to ~400 km scales, the subsequent effects of averaging to 3 h scales are negligible.

In both cases, the spatially averaged spectra of fractional contributions to 3-hourly rainfall peak at lower intensities than the satellite rainfall datasets. In N1024p, both spatial and temporal intermittency contribute to this underestimate. In N1024e, even the excessive rainfall amounts in isolated grid boxes are not sufficient to compensate for the large number of surrounding grid-boxes with no rainfall. However, for the WA region, the N1024e simulation does appear to represent the distribution of 3-hourly rainfall rather better than N1024p. In configurations with parametrized convection, some of the underestimate in the 3 h totals over the land regions is related to the poor diurnal cycle of rainfall (see e.g., Stratton and Stirling, 2012; Kendon et al., 2012), whereby deep convection starts and ends too early in the day and rainfall amounts in the evening and overnight are underestimated. The configuration with explicit convection, in common with other convection-permitting configurations (e.g., Hohenegger et al., 2008), has improved timing of this diurnal cycle (not shown), partly due to the unrealistic size of the grid-boxes used, which delay the start of deep convection but results in rainfall amounts that, once started, are very large and persist for a few hours. Similar analysis of a 4.5 km resolution MetUM configuration with explicit convection over Africa also shows an improved diurnal cycle of convection over land, but suggests that the extreme spatial intermittency is reduced as the grid-size decreases, and that the overall rainfall bias is smaller (Rachel Stratton, pers. comm.). Thus, our analysis demonstrates that the ASoP1 methods are able to identifying contrasting behavior of rainfall variability between simulations with parameterized and explicit convection. As noted previously, the grid-size used for this experiment is clearly unrealistic for explicit convection. Indeed, the use of a N1024 resolution for parametrized convection may also be questionable (Molinari and Dudek, 1992). Future work using these methods will investigate how these characteristics change as resolution is increased further towards the ~100 m scale.

## 5. Discussion and Conclusions

In order to have confidence in climate model projections of precipitation, it must be demonstrated that the modelled rainfall responds appropriately to changing atmospheric conditions on all scales. The ASoP1 methods designed by Klingaman et al. (2016) provide an additional tool for comparing and evaluating simulated rainfall variability between model configurations and with various observational datasets. Analysis of the spatial and temporal characteristics of rainfall in a set of parallel configurations of the MetUM-GA6 model using the ASoP1 methods has allowed several characteristics of tropical convection in these model configurations to be identified:

1. Precipitation produced by the convection parametrization on the native grid and time-step in MetUM-GA6 is both spatially and temporally intermittent, regardless of the horizontal resolution and time-step of the model, at least for the broad range of resolutions (20–200 km) and time-steps (5–20 min) considered here. This behaviour is caused by the choice of closure at GA6, in which the mass flux amplitude is set to depend on the CAPE detected in the grid-box, rather than the rate of atmospheric destabilisation. The resultant heating applied produces an inversion at the top of

the boundary layer on the next time-step that the diagnosis deems too strong to allow convection to initiate. It remains in this state until the inversion has been eroded by a combination of heating in the boundary layer, and large-scale ascent. This behaviour occurs immediately at the start of the simulation with no spin-up, regardless of grid size or time-step length. Klingaman et al. (2016) found similar behavior in two-day forecasts with an earlier MetUM version.

2. With parametrized convection, the fractional contributions to total precipitation from different intensities on the native grid and time-step are also largely insensitive to horizontal resolution and time-step in MetUM-GA6.

3. When the convection parametrization is switched off, albeit at an unrealistic resolution for explicit convection to be represented properly, the time-step precipitation becomes very persistent on scales up to the order of a few hours, but even more isolated on the grid-scale, likely due to the considerable dynamical forcing required to lift a 20 km $\times$ 13 km grid-box. Convective heating associated with explicit convection sets up significant ascent in the convecting column, which continues the destabilisation of the column, while adjacent columns experience descent and so convection is suppressed.

4. For MetUM-GA6 configurations with parametrized convection, spatial and temporal averaging to scales ~400 km and ~3 h reduces the spatial and temporal intermittency considerably. At these scales, the convection scheme starts to display sensitivity to the large-scale forcing, as the strength of this is what determines the frequency with which the convection scheme can be activated. However, MetUM-GA6 produces precipitation features that are too broad relative to the TRMM and CMORPH satellite-derived analyses.

5. For the MetUM-GA6 configuration with explicit convection used here, temporal averaging to scales ~3 h has little effect on the rainfall intensities, while spatial averaging to scales ~400 km has a very large effect, due to the large spatial intermittency.

6. Comparison of the model configurations' tropical precipitation variability on horizontal scales ~400 km and timescales from daily to 20 days (intraseasonal) shows no systematic difference in behaviour between the different resolutions. In all cases, the model tends to underestimate the amplitude of the intraseasonal variations (i.e. there are not enough drier days), over the ocean, at all resolutions.

7. The lack of intraseasonal variability contributes to an overall wet bias in some oceanic regions (e.g., the equatorial Indian Ocean, the western Pacific and South China), while underestimations of rainfall intensity on sub-daily and daily timescales in West Africa are associated with a climatological dry bias.

Attributing climatological biases in regional precipitation to deficiencies in model physical parametrizations remains a challenge for model developers. Such biases can have implications for weather and climate modelling on a wide range of temporal and spatial scales, from inhibiting moisture transport through intraseasonal propagation of convection (e.g., Bush et al., 2015; Kim et al., 2016) to contributing to uncertainty in projections of future tropical rainfall (e.g., Kent et al., 2015). By examining the behaviour of modelled tropical rainfall at a wide range of spatial and temporal scales, we can hope to shed light on the way in which such biases develop. Our results suggest that, in many regions, sub-seasonal tropical rainfall in the MetUM-GA6 configuration lacks variability on all but the smallest available temporal and spatial scales (i.e. the model time-

step and grid-scale). This suggests a lack of response from the convection parametrization to changing atmospheric conditions. Instead, at the time-step and grid-scale, the spatial and temporal intermittency appears to be quasi-random, much like the MetUM-GA3 configuration analysed by Klingaman et al. (2016). Such analysis provides information to model developers which should help to inform the future direction of parametrization development.

The apparent lack of sensitivity to horizontal resolution is, at first sight, in contrast with other model studies which suggest an improvement in rainfall characteristics as horizontal resolution is increased (e.g., Wehner et al., 2010; Kopparla et al., 2013; Prein et al., 2013; Tripathi and Dominguez, 2013). However, several of these studies compare the results on the native grid of each model with observations at resolutions that are often higher than in any of the model configurations. This will, naturally, highlight the improved representation of the natural spatial variability of rainfall arising from local dynamical

gradients, orography, etc., in higher resolution models. Indeed, Prein et al. (2013) comment that "The major advantages of high-resolution simulations are found for small scales" and that, at scales above ~100 km, their higher resolution runs show only "small advantages" over their lower resolution runs. However, Kopparla et al. (2013) point out that such comparisons differ between regions.

     Furthermore, the use of daily mean values in most of these studies hides issues with sub-daily variability such as spatial

and temporal intermittency and a poor diurnal cycle. We acknowledge that many of the characteristics we highlight in our study may be particular to the MetUM-GA6 configuration and its convection parametrization. However, Klingaman et al. (2016) showed that there are other models, with a wide variety of horizontal resolutions, which also show spatial and intermittency in time-step tropical rainfall. We hope that our results will encourage similar systematic studies of the effects of horizontal resolution in other models. Our finding that the effects of switching off the convection parametrization and

allowing explicit convection, albeit on unrealistic spatial scales, has a much more marked impact on the tropical rainfall characteristics, motivates further study on the sensitivity to the convection parametrization itself. A recent study by Jin et al. (2016) suggests that resolution sensitivity in the diurnal cycle of rainfall over China simulated by the Weather Research and Forecasting (WRF) model is strongly related to the increasing contribution from non-convective rainfall, while the contribution from convective precipitation remains similar unless the convection parametrization is altered.

Finally, the increasing number of studies using convection-permitting resolutions with grid-lengths of a few kilometres (e.g., Fosser et al., 2015; Kendon et al., 2012; Prein et al., 2013) suggest that, while such models do exhibit improved sub-daily characteristics and diurnal cycles of rainfall, problems remain with excessive rainfall rates and too-persistent precipitation events at these resolutions. It is clear that similar analyses of tropical rainfall characteristics in convection-permitting models at resolutions <~1 km would be enlightening and of significant use in model development.

**Data and code availability**

The source code for the model used in this study, MetUM, is free to use. To apply for a licence for MetUM go to http://www.metoffice.gov.uk/research/collaboration/um-collaboration. The availability of the ASoP1 diagnostics package is

detailed in Klingaman et al. (2016). MetUM-GA6 model data are archived at the Met Office, and are currently available to UM partners. TRMM 3B42 version 7A data can be obtained from http://disc.sci.gsfc.nasa.gov/TRMM. CMORPH version 1.0 data can be obtained from ftp://ftp.cpc.ncep.noaa.gov/precip/global_CMORPH/3-hourly_025deg.

## Author contributions

N. Klingaman analysed the spatial and temporal intermittency and G. Martin and A. Moise analysed the spectral characteristics of the rainfall data. G. Martin and N. Klingaman wrote the manuscript with input from A. Moise.

## Competing interests

The authors declare that they have no conflict of interest.

## Acknowledgements

G. Martin was supported by the Joint DECC/DEFRA Met Office Hadley Centre Climate Programme (GA01101). N. Klingaman was supported by an Independent Research Fellowship from the UK Natural Environment Research Council (NE/L010976/1). A. Moise was supported by funding from the Australian Climate Change Science Program. The authors are grateful to the model development teams at the Met Office who ran the MetUM-GA6 simulations as part of the Global Atmosphere 6.0 development process, and to Alison Stirling for helpful comments on the manuscript.

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

**Table 1. For each MetUM-GA6.0 simulation: the name of the simulation, the horizontal resolution in degrees (to the nearest 0.01°) and the equivalent in km at the equator, the time-step, the largest domain over which data were extracted, the representation of deep convection (either a sub-grid-scale parametrization or entirely "explicit" convection) and the years of daily(time-step) data analysed. Note that data are limited to June, July, August, September of any given year. Models are listed in order of decreasing**

5    **horizontal grid spacing, as in the figures.**

| Name | Lon° x Lat° (km) | Time-step minutes | Available data domain | Deep convection | Years analysed for daily (time-step) data |
|------|------------------|-------------------|-----------------------|-----------------|-------------------------------------------|
| N96 | 1.88° x 1.25° (210 x 139) | 20 | 20°S – 40°N, 20°W – 160°E | parametrized | 1982-2008 (1990) |
| N216 | 0.83° x 0.56° (92 x 62) | 15 | 20°S – 40°N, 20°W – 160°E | parametrized | 1982-2008 (1990) |
| N512 | 0.35° x 0.23° (39 x 26) | 10 | 20°S – 40°N, 20°W – 160°E | parametrized | 1982-1990 (2007) |
| N1024p | 0.18° x 0.12° (20 x 13) | 5 | 0° – 20°N, 130° – 160° E | parametrized | (2005) |
| N1024p | 0.18° x 0.12° (20 x 13) | 5 | 8° – 17°N, 0° – 10° E | parametrized | (2005) |
| N1024e | 0.18° x 0.12° (20 x 13) | 5 | 0° – 20°N, 130° – 160° E | explicit | (2005) |
| N1024e | 0.18° x 0.12° (20 x 13) | 5 | 8° – 17°N, 0° – 10° E | explicit | (2005) |

**Table 2:** For each MetUM-GA6 resolution, as well as TRMM and CMORPH: the analysis region used; the dimensions of each region in native grid-boxes; the number of model time-steps in 3 hours; the number of native grid-boxes in an N48 grid-box (rounded to the nearest whole grid-box); the number of $7 \times 7$ native-grid-box regions in the analysis domain; and the number of "equal-area" 1700 km and 600 km regions in the analysis domain, with the dimensions of the regions (in native grid-boxes) shown in parentheses.

| Dataset | Region | Size (nx × ny) | $\Delta t$ in 3 h | # of boxes in N48 | # of 7 × 7 regions | # of 1700 km regions (nx × ny) | # of 600 km regions (nx × ny) |
|---|---|---|---|---|---|---|---|
| N96 | EQ | 53 × 16 | 9 | 4 | 14 | 6 (8 × 12) | N/A |
| N216 | EQ | 120 × 36 | 12 | 20 | 85 | 6 (18 × 27) | N/A |
| N512 | EQ | 284 × 86 | 18 | 113 | 480 | 6 (43 × 64) | N/A |
| N512 | WP | 85 × 85 | 18 | 113 | 144 | N/A | 15 (15 × 23) |
| N1024p | WP | 170 × 171 | 36 | 455 | 576 | N/A | 15 (30 × 46) |
| N1024e | WP | 170 × 171 | 36 | 455 | 576 | N/A | 15 (30 × 46) |
| N48 averaged | EQ | 28 × 9 | N/A | 1 | 4 | 6 (4 × 6) | N/A |
| N48 averaged | WP | 9 × 9 | N/A | 1 | 1 | 2 (4 × 6) | N/A |
| TRMM | EQ | 400 × 80 | 1 | 150 | 627 | 6 (51 × 51) | N/A |
| CMORPH | EQ | 400 × 80 | 1 | 150 | 627 | 6 (51 × 51) | N/A |
| CMORPH | WP | | 1 | 150 | 187 | N/A | 15 (22x22) |

**Table 3: Summary metrics of spatial and temporal coherence in precipitation using time-step and 3 h data on the native horizontal grid and averaged to the N48 (3.75°x2.5°) grid. Positive values indicate that coherence is more common than intermittency. Higher positive (negative) magnitudes indicate stronger coherence (intermittency). The time-step column is marked "N/A" for TRMM and CMORPH because these datasets exist only as 3-hr values.**

| Dataset | Region | Spatial coherence | | | | Temporal coherence | | | |
| --- | --- | --- | --- | --- | --- | --- | --- | --- | --- |
| | | Native grid | | N48 grid | | Native grid | | N48 grid | |
| | | Time-step | 3 h | Time-step | 3 h | Time-step | 3 h | Time-step | 3 h |
| N96 | EQ | 0.27 | 0.58 | 0.32 | 0.41 | -0.01 | 0.68 | 0.32 | 0.74 |
| N216 | EQ | 0.29 | 0.65 | 0.39 | 0.44 | -0.09 | 0.74 | 0.62 | 0.75 |
| N512 | EQ | 0.33 | 0.83 | 0.42 | 0.44 | -0.03 | 0.60 | 0.80 | 0.75 |
| N512 | WP | 0.27 | 0.86 | 0.42 | 0.45 | -0.15 | 0.63 | 0.78 | 0.78 |
| N1024p | WP | 0.28 | 0.92 | 0.34 | 0.35 | -0.11 | 0.61 | 0.89 | 0.79 |
| N1024e | WP | 0.68 | 0.72 | 0.16 | 0.17 | 0.91 | 0.38 | 0.98 | 0.63 |
| TRMM | EQ | N/A | 0.72 | N/A | 0.34 | N/A | 0.33 | N/A | 0.58 |
| CMORPH | EQ | N/A | 0.76 | N/A | 0.37 | N/A | 0.43 | N/A | 0.66 |
| CMORPH | WP | N/A | 0.80 | N/A | 0.42 | N/A | 0.45 | N/A | 0.69 |

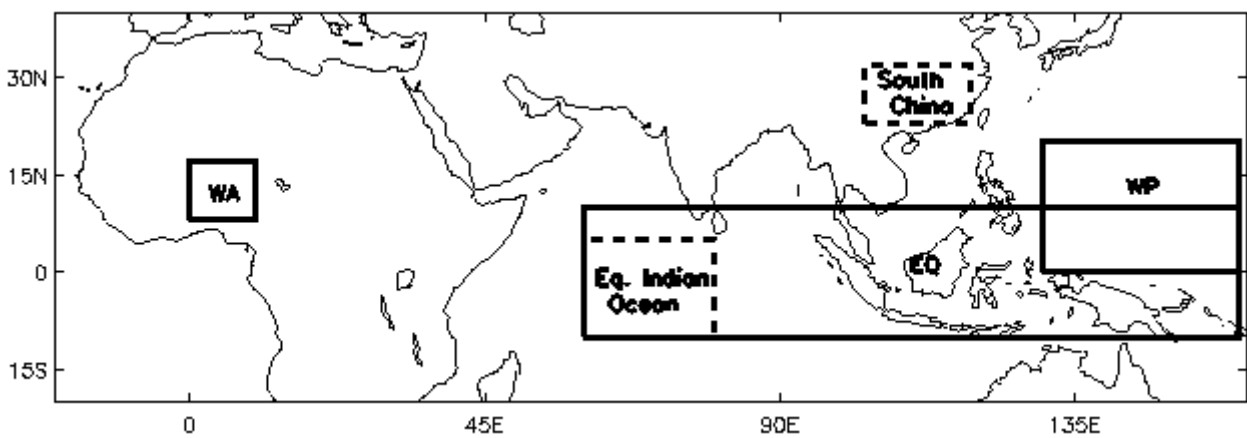

**Figure 1. Map illustrating the regions used in this study. "WA": West Africa; "WP" West Pacific; "EQ": Equatorial region. See text for further details.**

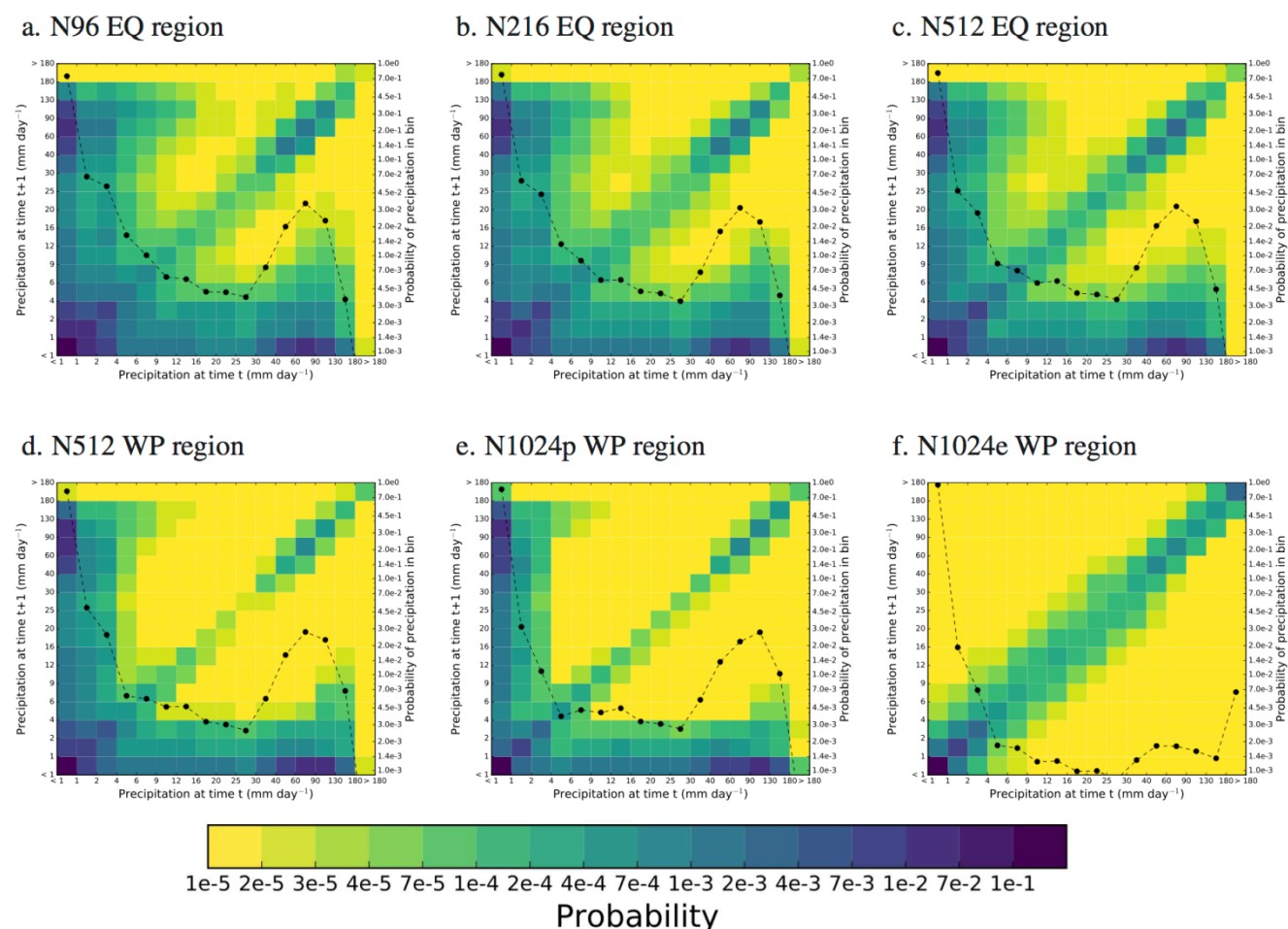

**Figure 2. For each MetUM GA6 configuration in Table 1, the filled blocks show the two-dimensional histogram of binned rain rates (in mm day$^{-1}$) on consecutive time-steps at the same grid box, aggregated over all grid boxes; the dashed line shows the one-dimensional histogram of binned precipitation, using the right-hand vertical axis. Bins were chosen qualitatively such that 3-hourly TRMM analyses over the EQ region have an approximately uniform distribution for rain rates greater than 1 mm day$^{-1}$. Note the logarithmic colour scale.**

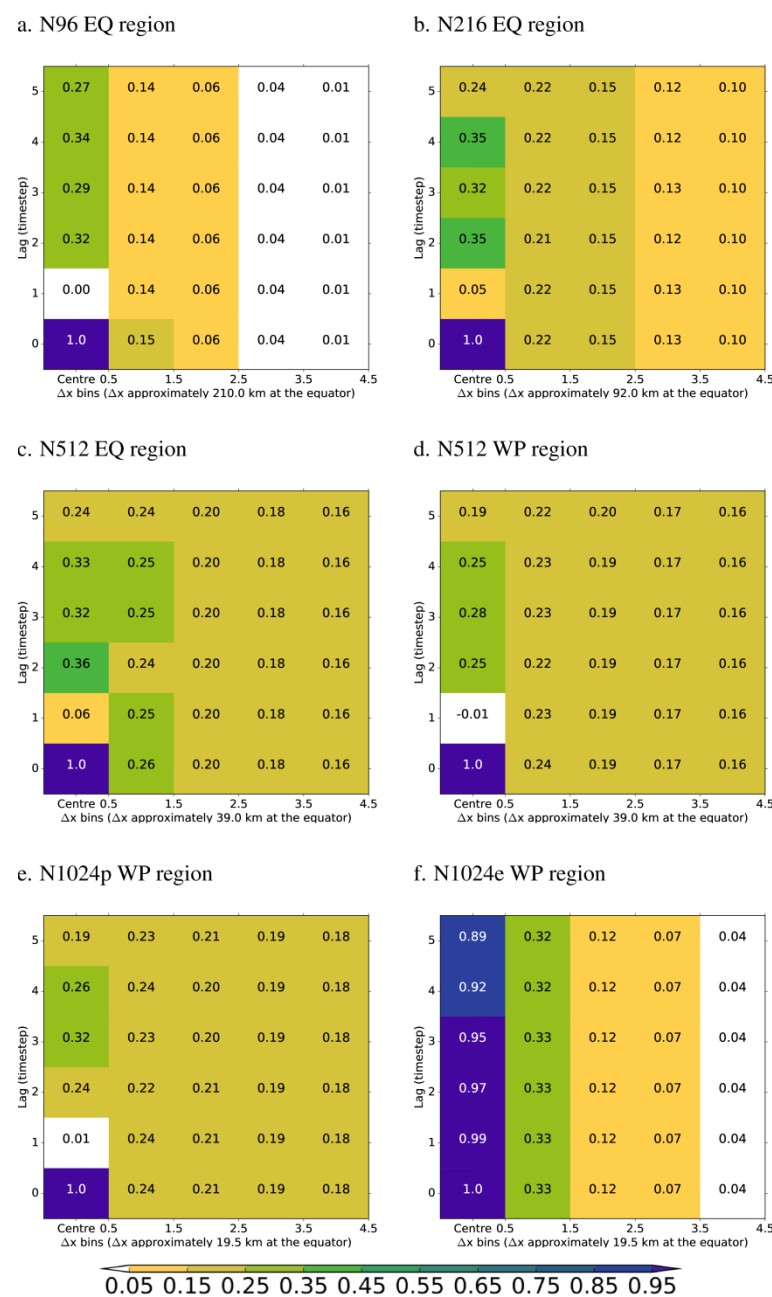

**Figure 3. For each model and using time-step precipitation on the native horizontal grid, filled boxes and numbers show the lagged correlations between the central grid-box in each 7x7 sub-region and grid-boxes within each range of distance on the horizontal axis (in units of the longitudinal grid spacing at the equator, $\Delta x$) away from the central point, averaged over all 7x7 regions. 'Centre' denotes the auto-correlation at the central grid-box.**

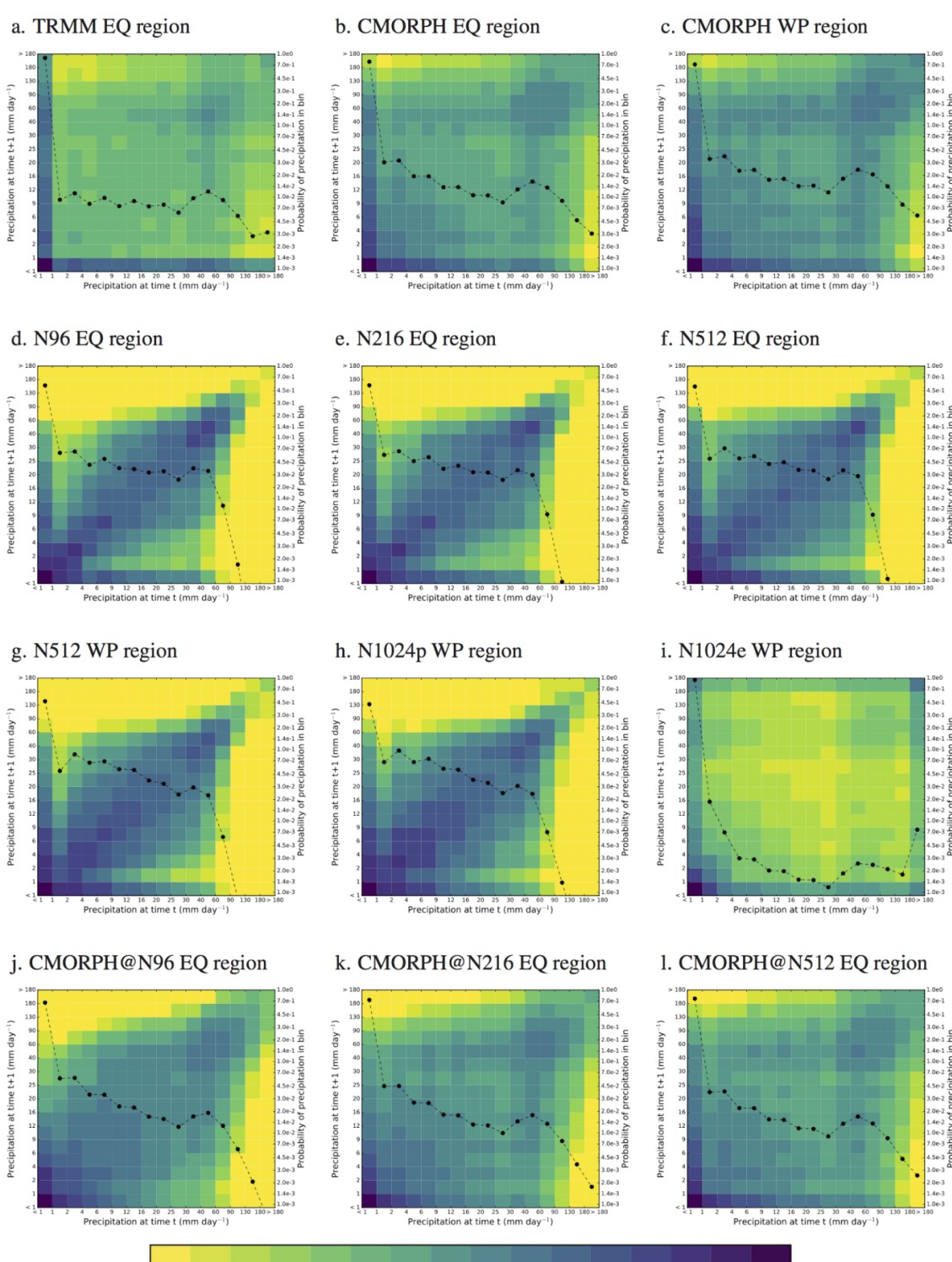

**Figure 4. As in Fig. 2, but using 3 h mean rain rates instead of time-step rain rates, retaining the native horizontal grids. Panels a-c show 3-hourly CMORPH and TRMM data for JJAS 2005, using their native grids. CMORPH is shown for both EQ and WP to demonstrate the similarity between the regions. Panels j-l show CMORPH averaged to the N96, N216 and N512 MetUM resolutions, respectively, over the EQ region, to compare with panels d-f.**

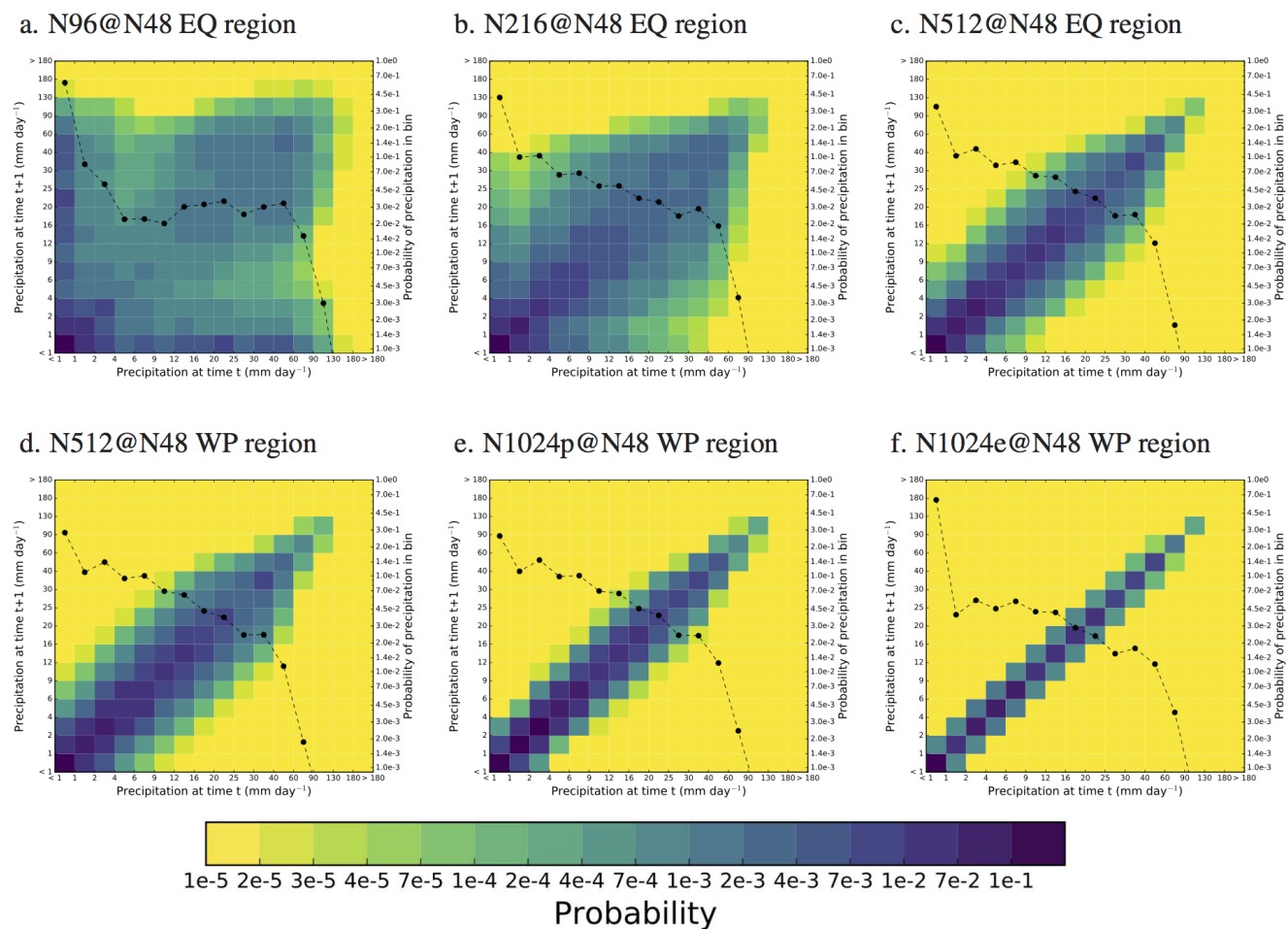

**Figure 5: As in Fig. 2, but using time-step rain rates that were first spatially averaged to a 3.75° ✕ 2.5° horizontal grid (MetUM N48 resolution).**

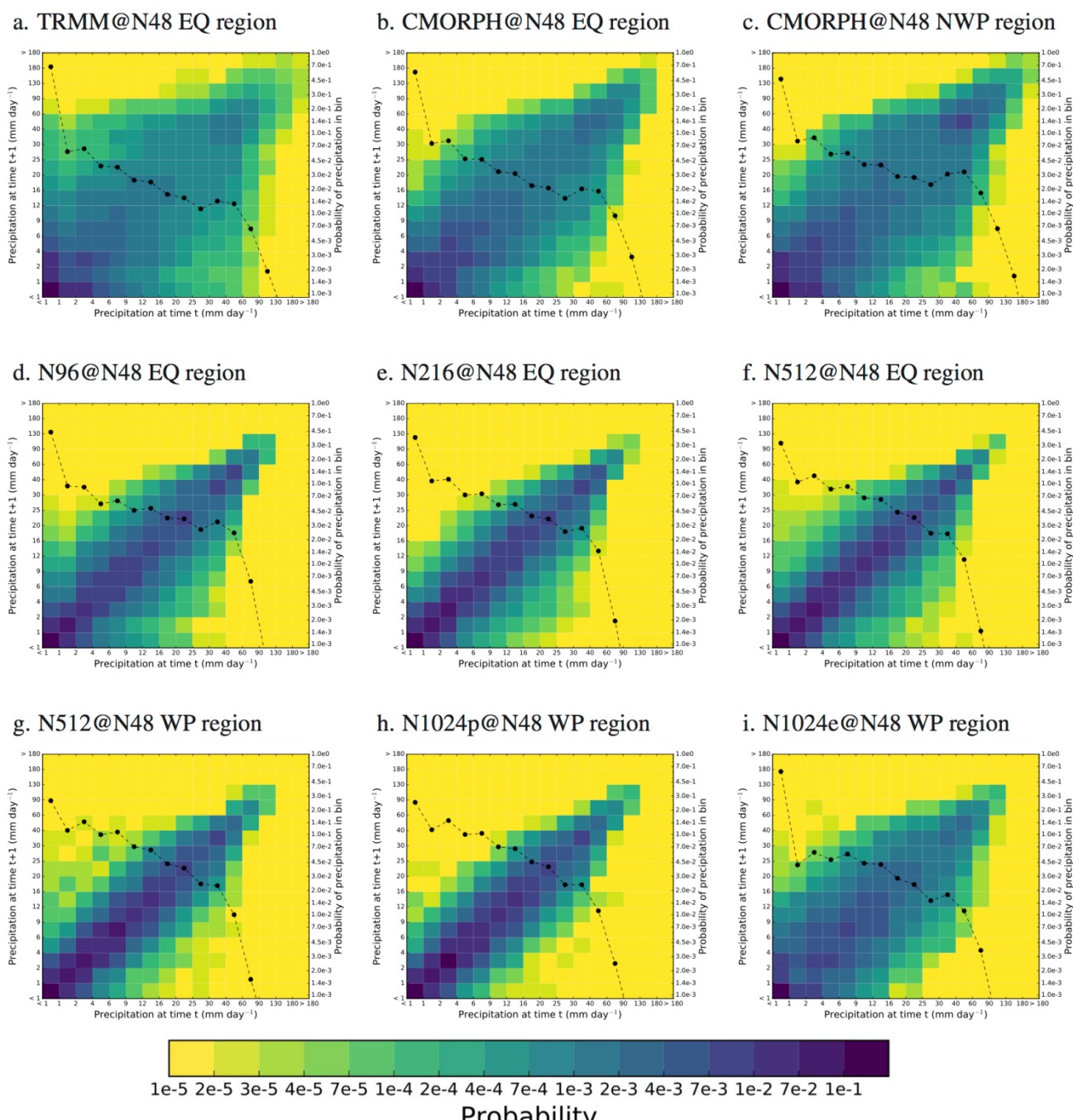

**Figure 6.** As in Fig. 2, but using 3 h mean rain rates interpolated the 3.75° ✕ 2.5° grid (MetUM N48 horizontal resolution).

a. Spatial – timestep, native grid

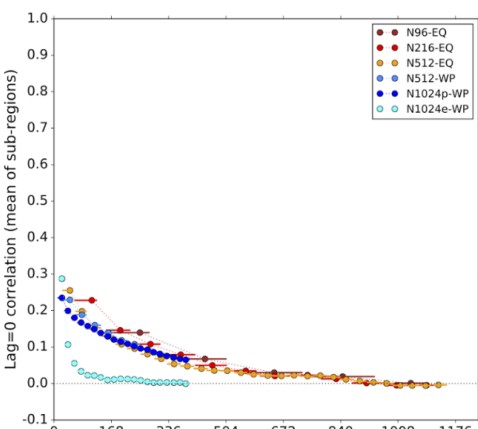

b. Spatial – timestep, N48 (3.75°×2.5°) grid

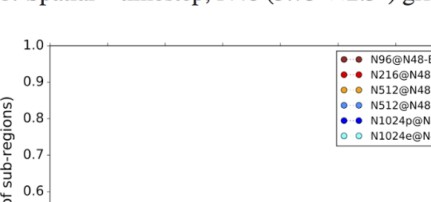
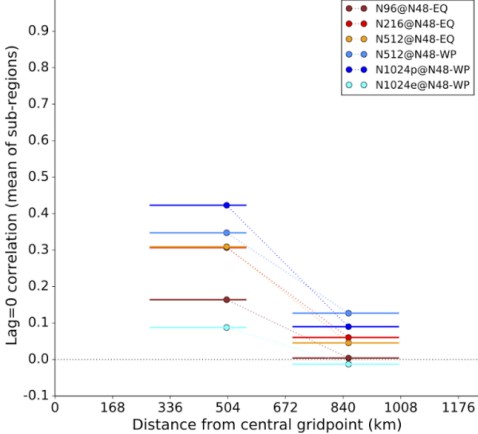

c. Spatial – 3-hr means, native grid

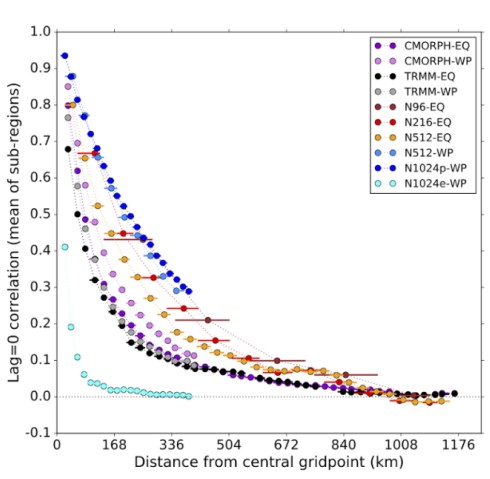

d. Spatial – 3-hr means, N48 (3.75°×2.5°) grid

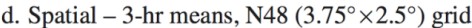
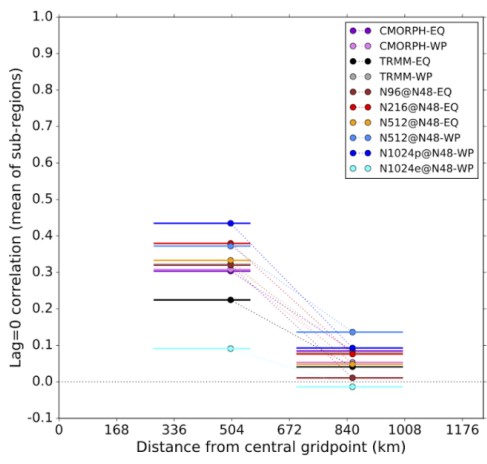

e. Time – timestep, native grid

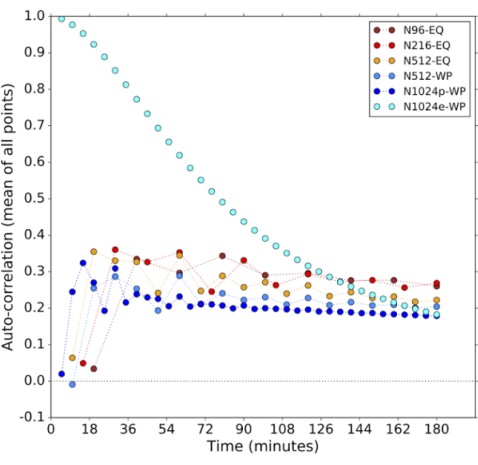

f. Time – timestep, N48 (3.75°×2.5°) grid

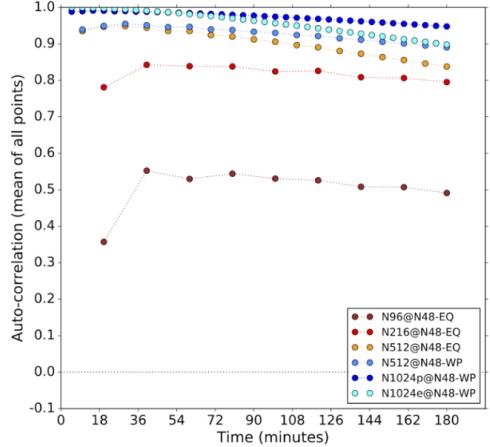

**Figure 7. (a to d) A measure of the spatial scale of precipitation, computed by dividing the domain into equal-area regions and calculating the lag-0 correlations between the central grid-box and grid-boxes within each distance bin (which are $\Delta x$ wide, starting from $0.5\Delta x$) away from the central grid-box, then averaging correlations over all regions in the domain, using: (a) time-step rain rates on the model configurations' native horizontal grid, (b) time-step rain rates averaged to the N48 horizontal grid (c) 3-hourly rain rates on the native horizontal grid, and (d) 3-hourly rain rates on the N48 horizontal grid; (e, f) a measure of the temporal scale of precipitation, computed as the auto-correlation of precipitation, averaged over all boxes in the domain, using (e) time-step rain rates on the models' native horizontal grid and (f) time-step rain rates on the N48 horizontal grid. The horizontal lines in (a-d) show the range of distances spanned by each distance bin; the filled circle is placed at the median distance. For clarity, we omit the correlations for zero distance and zero lag, which are 1.0 by definition. In the legends, "-EQ" refers to the EQ analysis domain and "-WP" to the WP analysis domain; "@N48" refers to data averaged to the N48 horizontal grid.**

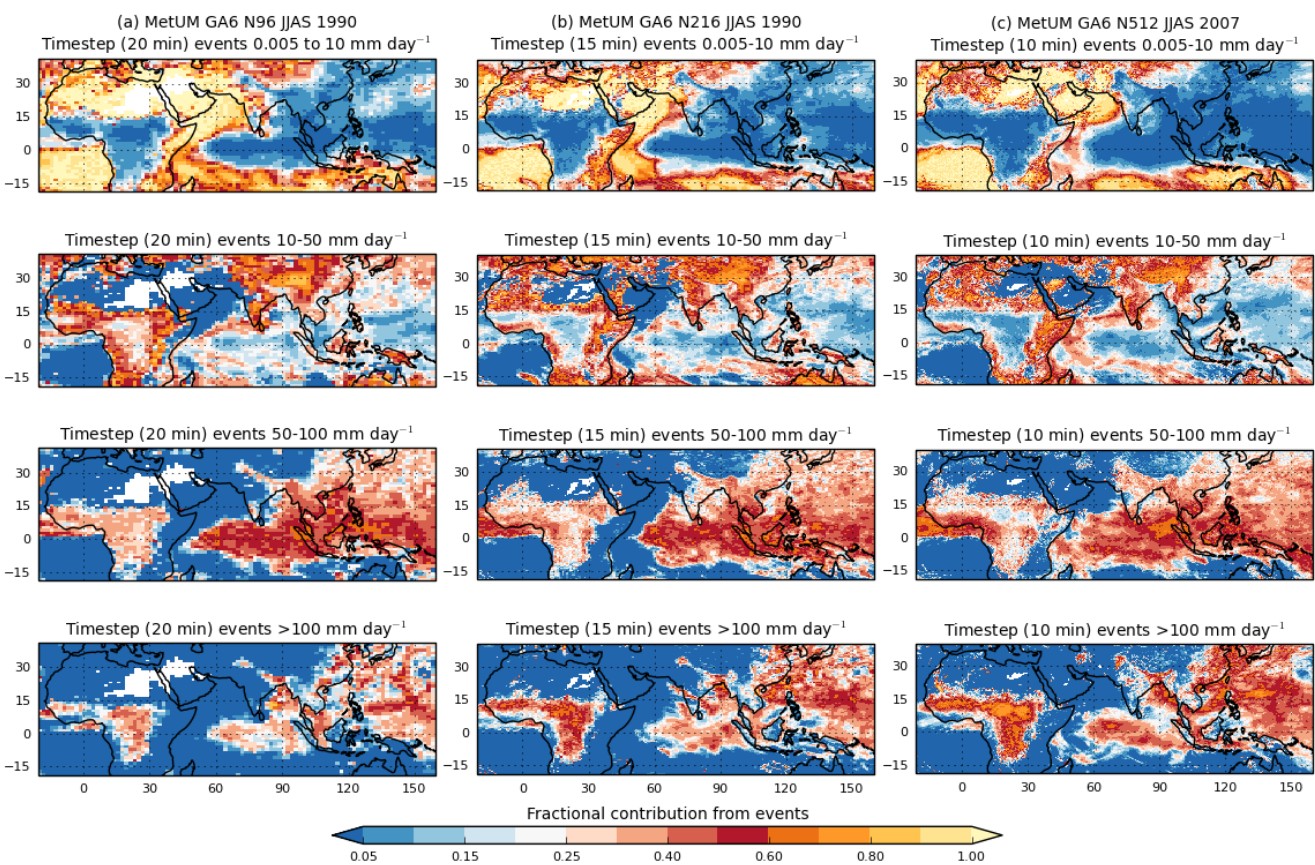

**Figure 8. Spectral maps of time-step precipitation in JJAS from three MetUM GA6 configurations: (a) N96, (b) N216 and (c) N512. Data are analysed on each configuration's native grid. For each panel, the fractional contributions from all bins within the given intensity range are summed at each grid-box. Time-step lengths and years analysed are shown in Table 1.**

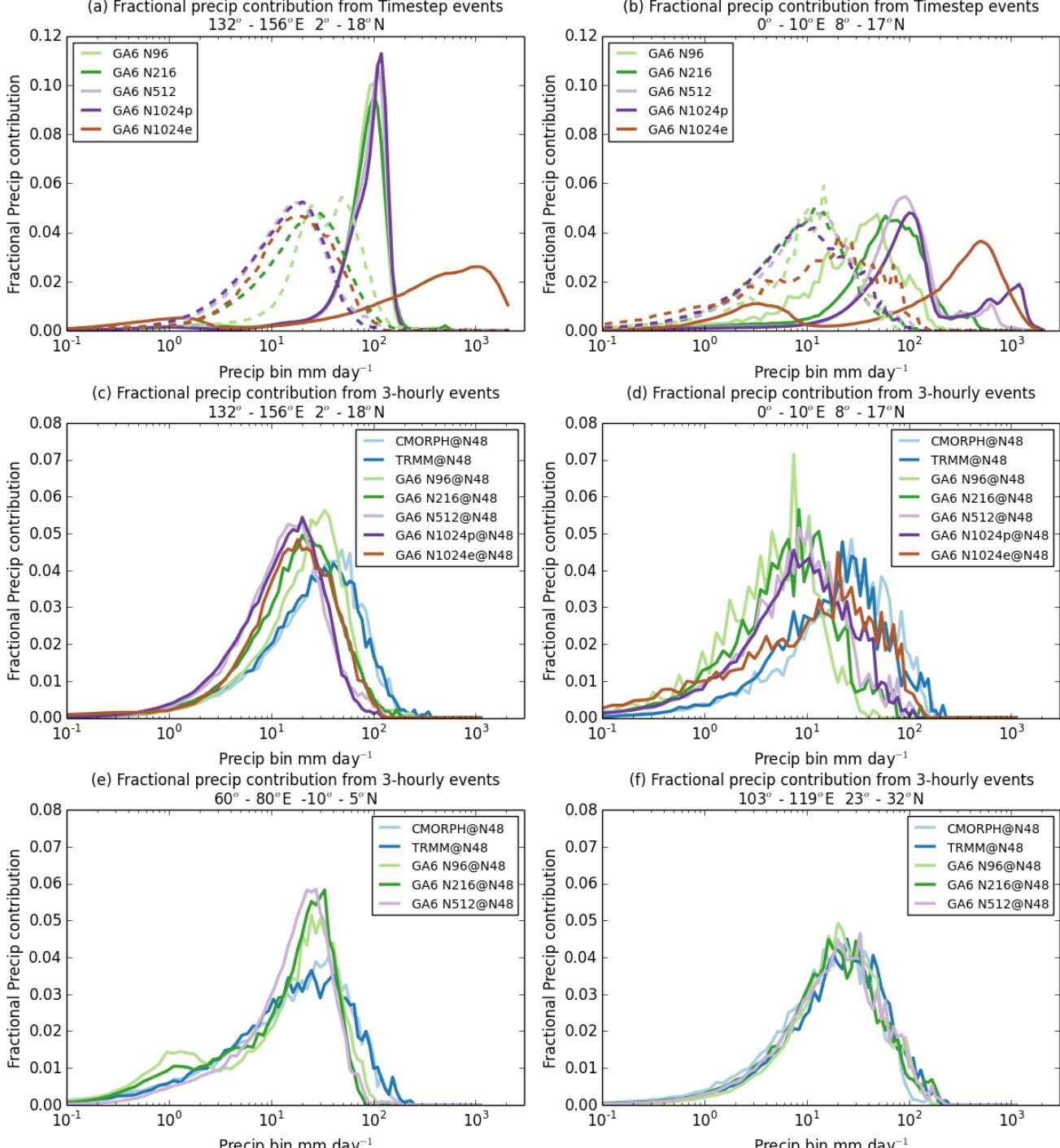

**Figure 9. (a and b) Precipitation spectra averaged over different regions for the five model configurations. Solid lines show spectra at native resolution and time-step while dashed lines show the spectra from each configuration when precipitation data are averaged to the N48 grid: (a) "WP" domain; (b) "WA" domain. (c to f) Precipitation spectra averaged to the N48 grid and 3 h timescale, for: (c) "WP" domain; (d) "WA" domain; (e) Equatorial Indian Ocean (60°–80°E, 10°S–5°N); (f) South China (103°–119°E, 23°–32°N). N1024 data are not available for the latter two regions. Also included in each panel are results from the CMORPH and TRMM satellite-based rainfall analyses averaged to the same spatial and temporal scale.**

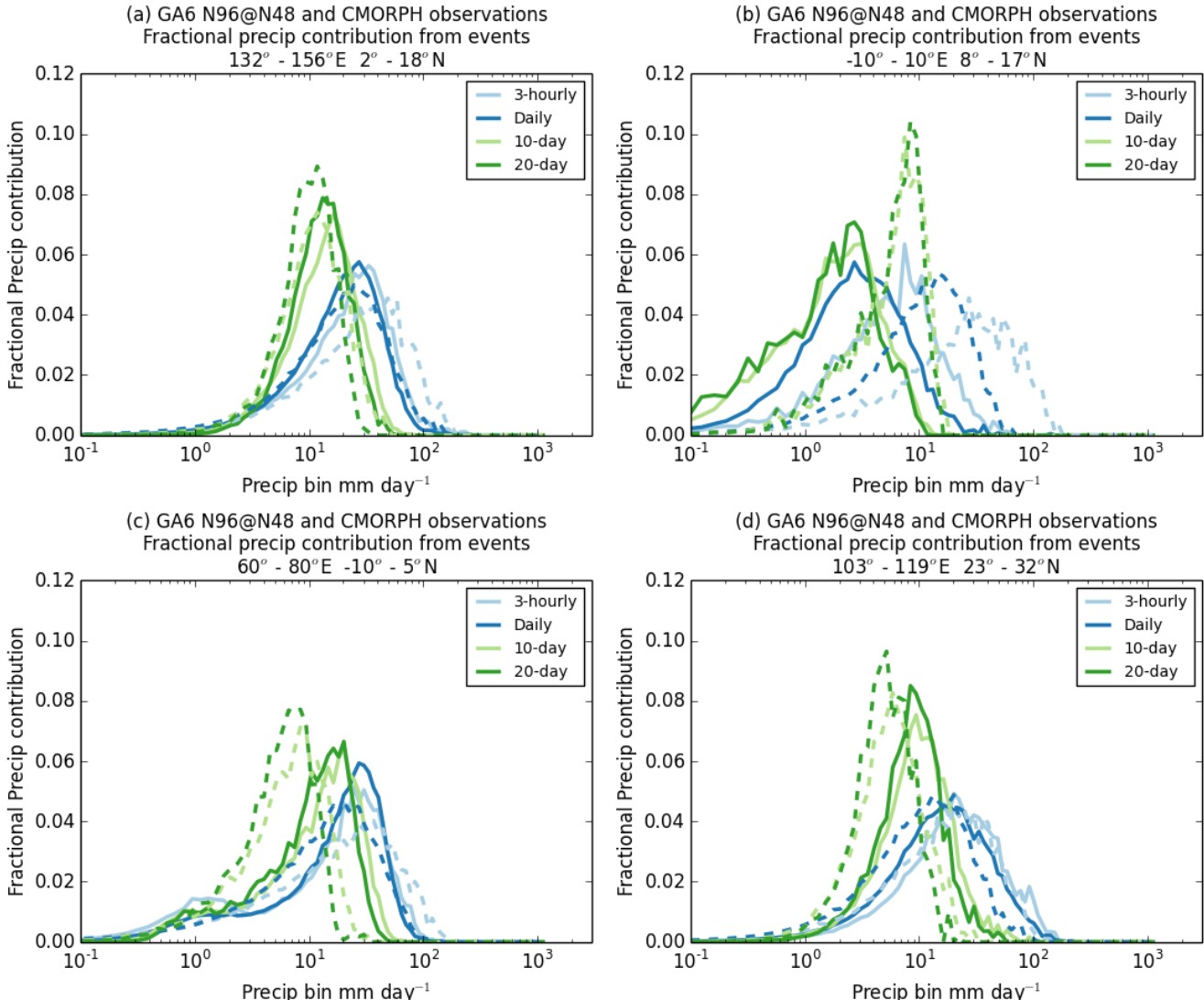

**Figure 10. Spectra of 3-hourly, daily, 10 day and 20 day rainfall totals (in mm day⁻¹) from the MetUM-GA6 N96 configuration (solid lines), averaged over four regions: (a) "WP" domain; (b) "WA" domain; (c) Equatorial Indian Ocean (10°S–5°N, 60°–80°E); (d) South China (23°–32°N, 103°–119°E). Corresponding spectra from CMORPH are shown by the dashed lines. Rainfall data were first averaged to the N48 grid in both cases.**

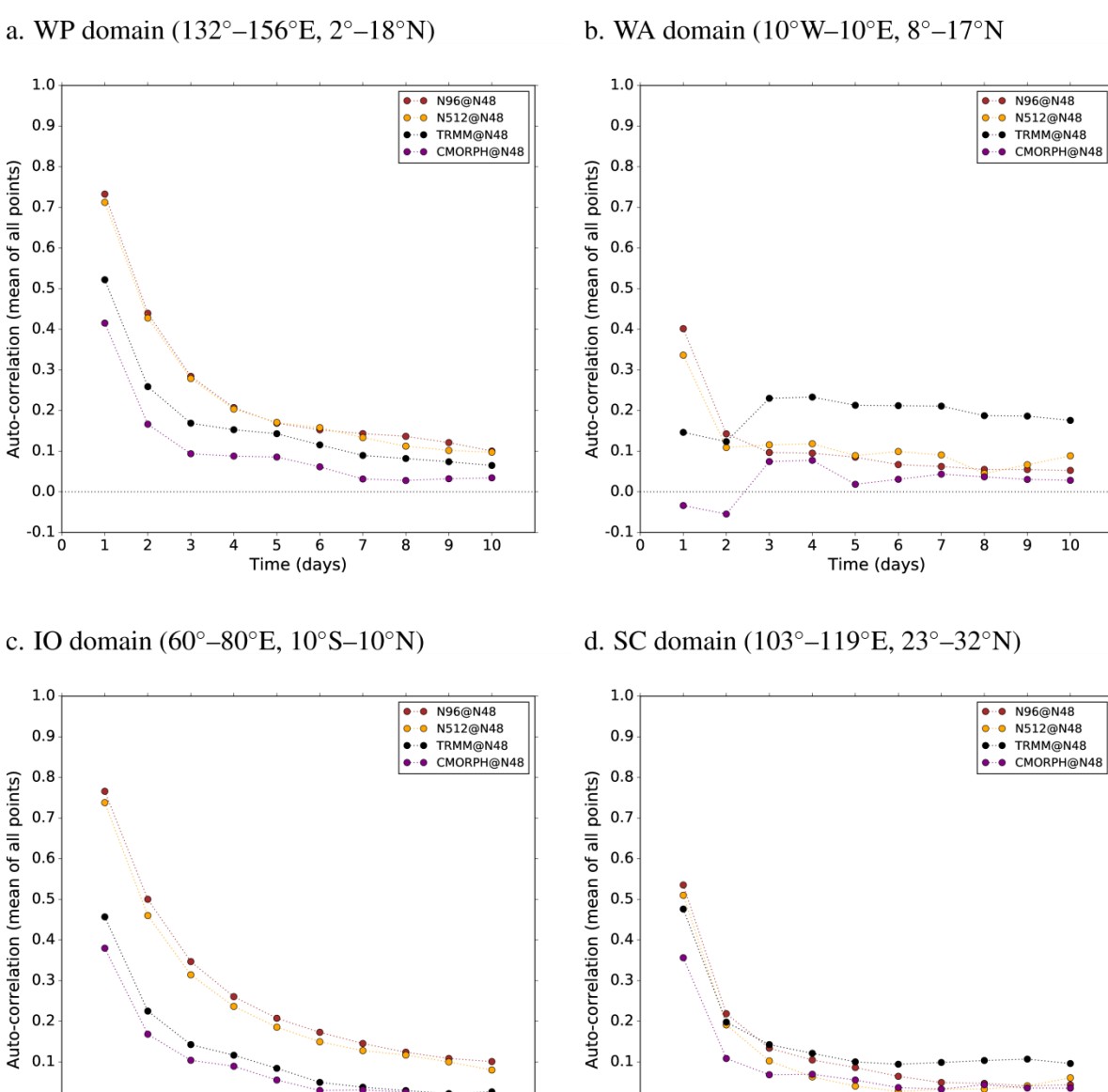

**Figure 11. Auto-correlations of precipitation at different time-lags, computed using daily precipitation on the N48 horizontal grid, averaged over the four regions shown in Figure 10.**

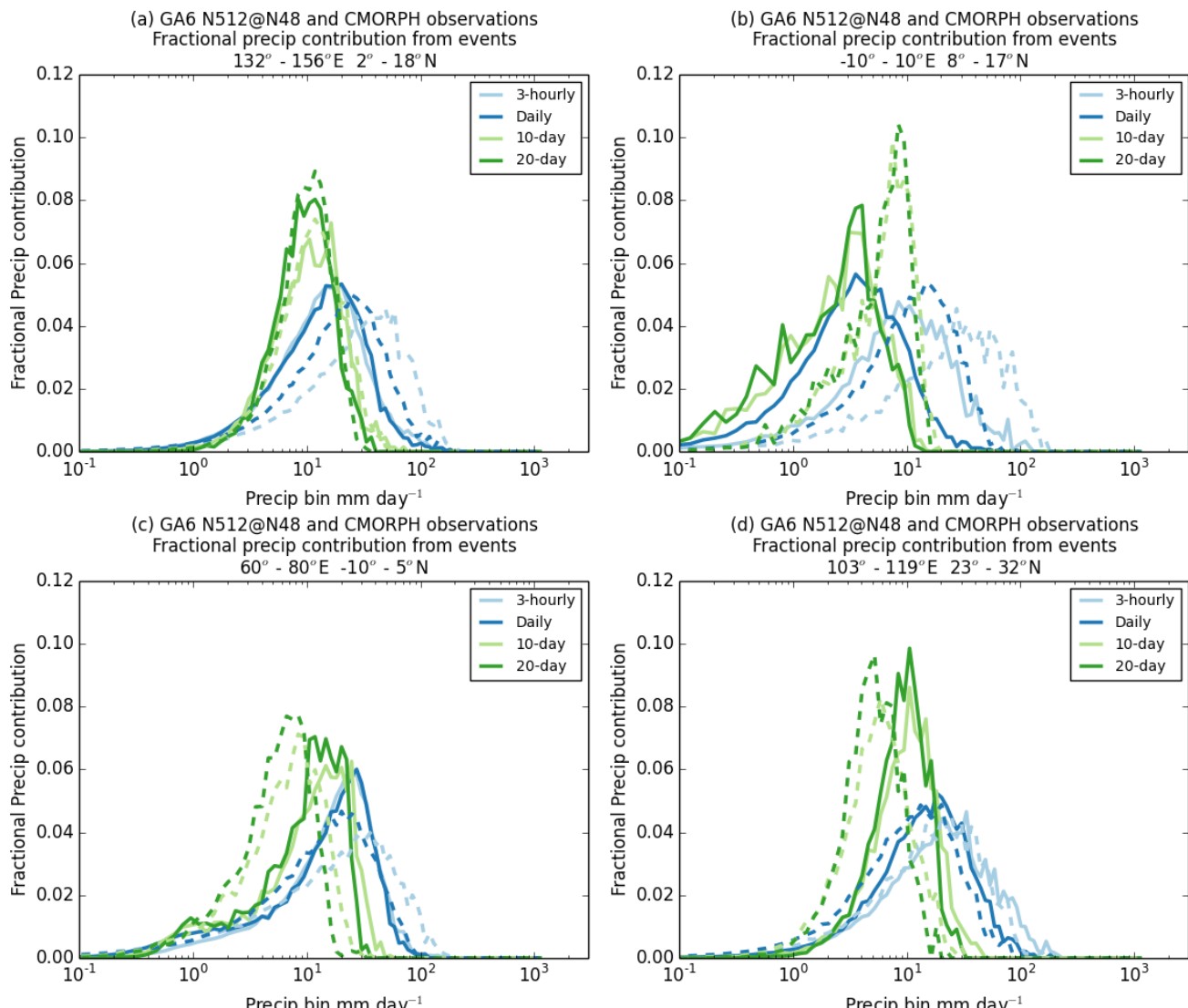

**Figure 12. As Fig. 10 but for the MetUM-GA6 N512 configuration.**

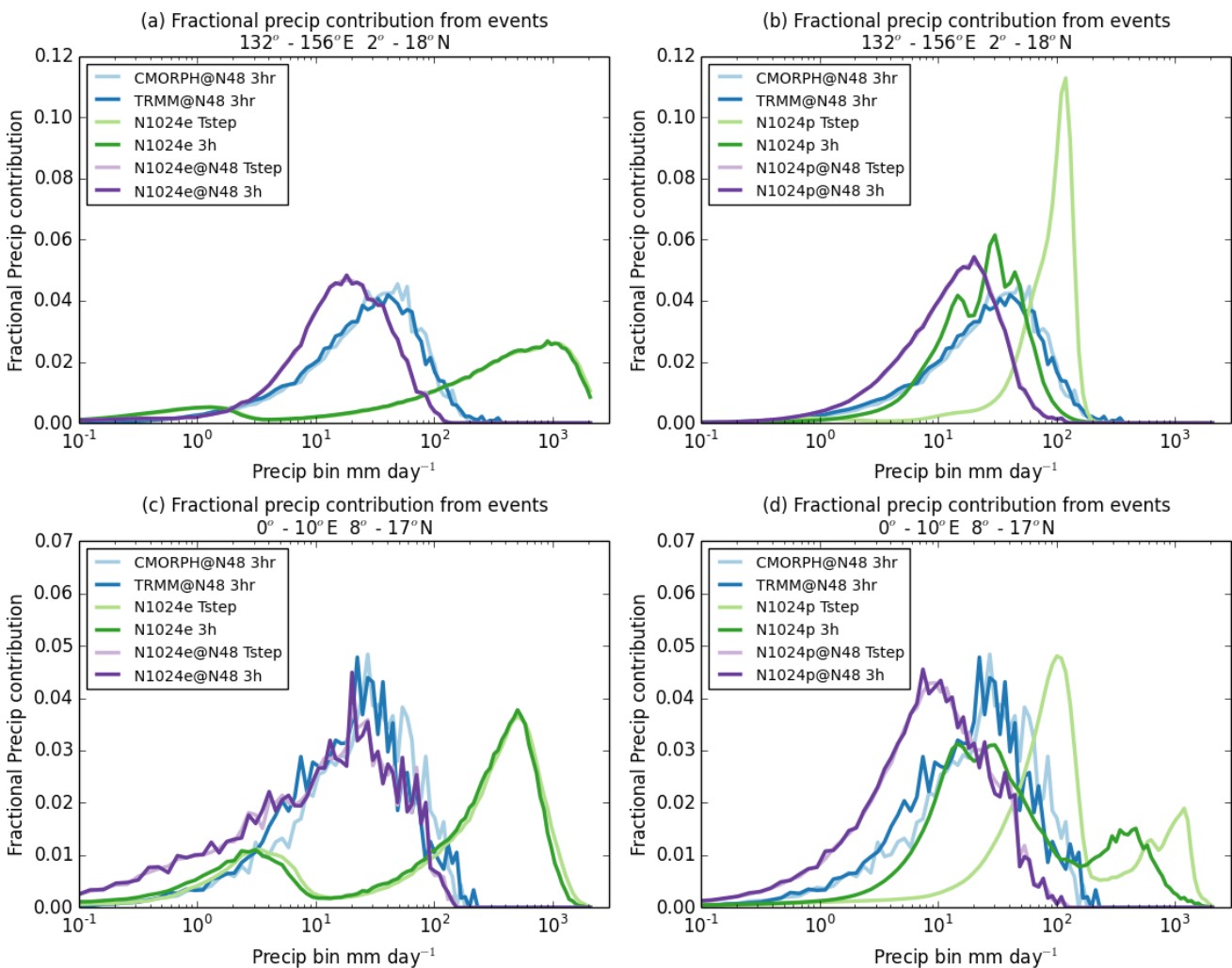

**Figure 13.** Spectra of time-step (5 min) and 3-hourly rainfall contributions, at native resolution and averaged to the N48 grid, from (left) MetUM-GA6 N1024e and (right) N1024p configurations, averaged over (top) the "WP" domain and (bottom) the "WA" domain.

