# Peer review of "Connecting spatial and temporal scales of tropical precipitation in observations and the MetUM-GA6"

_Geoscientific Model Development, 2016_

## Referee Comment (RC1) · Anonymous Referee #1 · 26 Oct 2016

General comments

This study is both concise and informative in its overview of how the intermittent behavior of parameterized precipitation is insensitive to horizontal resolution. The introduction references the appropriate literature and the methods section is clearly written (though it is easy to get bogged down with all the different geographical domains and scales). The authors' insight into how the spatial and temporal variability of parameterized precipitation influences both wet and dry biases in the model is particularly interesting (Figure 9 is wonderful), as these biases may be at the heart of the models' inability to properly represent tropical convection. Focusing on the variability of time-step precipitation in the context of extended/climate forecasts is an interesting

topic, though the authors themselves do not draw many connections between the intermittency of this short-timescale precipitation to the larger scales. The inclusion of an experiment without convective parameterization acts as a "tease" of (hopefully) future work done at higher resolution, though some interesting results were found here. Overall, the paper is well-written and presents a clever usage of the ASoP1 techniques to process a large volume of model output. Recommendation: Accept after minor alterations.

Specific comments

- While it is mentioned in your introduction that the representation of rainfall on short timescales is important to longer timescales (because of the biases they incite; Kendon et al. 2014), there is limited discussion of this topic thereafter. I think your hypothesis attributing the mean-state precipitation biases to issues with intraseasonal variability and sub-daily variability is one of the main takeaways of the paper; therefore, it would be good to include (perhaps in the discussion section) a paragraph highlighting the impact of these biases on the larger scales (e.g., a wet bias in the West Pacific may inhibit MJO propagation).

- You noted that many have observed that model biases develop fairly early on in climate simulations. Was this the case with your experiment? Did you observe any sort of spin-up time for the precipitation intermittency? Any note on the lead-dependence of the models' representation of precipitation would be a nice addition.

- In section 4.2, you only allude to intraseasonal variability (or the lack thereof) via proxy; that is, we know that if longer and longer temporal averages pull the precipitation spectra to smaller and smaller values, there must be variability on those longer timescales. A power spectral analysis of the observations vs the model, or anything directly showing this intraseasonal variance discrepancy between CMORPH and MetUM, would help drive this point home.

Technical corrections

- P4, L17-20: You use different grammar each time you introduce one of your three subdomains (see the "hereafter" bits) - Include a figure highlighting your subdomains and how they are divided up for section 3.5. - P5, L28-32: Add a figure or reference to illustrate/support your explanation of how the all-or-nothing nature of the convective parameterization is the cause of the precipitation intermittency. - P6, L30: "of this is what" to "thereof" - P7, L2: Did you look at the model output to see if this was the case? Are the differences in precipitation intermittency observable in the raw model output? - P8, L27: "(Fig. 6)" to "(Fig. 6f)" - P10, L7-9: These first two sentences of the paragraph should be earlier in the paper, as this is not the first time the data was averaged to N48 and 3-hourly to compare to CMORPH and TRMM (you did this in Figure 5 as well). - P10, L24-25: "indicates variability at the longer timescale" is awkward. Perhaps "is due to variability at longer timescales"
* * *

---

## Referee Comment (RC2) · Anonymous Referee #2 · 27 Oct 2016

General Comments

This paper presents a useful study on the behavior of tropical precipitation in the MetUM-GA6 model and the sensitivity to grid spacing. The introduction does a nice job setting up motivation for the project and includes a concise synthesis of prior work related to precipitation modeling. The methods are outlined clearly; however, more justification for the years chosen for analysis could be included. In the results sections, the figures and accompanying text clearly communicate the results; claims are backed up with reasonable explanations and limitations/caveats are noted throughout. The presentation of different types of analyses helps bring together the results of the paper, the main take-away being that in this model, precipitation characteristics are largely

unaffected by changing resolutions. Recommendation: Minor Revisions

Specific Comments

-Discussion of Table 1 on P4 mentions several different geographical domains – a figure outlining all domains used throughout the study would be useful.

-It is mentioned in the Methods discussion that there is little sensitivity in the year chosen for the time-step analysis (P4, L14-15), but is there any justification for why you chose the years you did? For instance, the simulation years for N512 noted in Table 1 are 1982-1990, but 2007 is used for the time-step analysis.

-The discussion of Fig. 1 talks about consistent intermittency between resolutions – this is qualitatively true, but difference PDFs, possibly between the highest and lowest resolutions, or some statistical significance testing could help show this more quantitatively.

-The end of section 3.4 discusses how the explicit convection results compare best to CMORPH/TRMM – consider mentioning that possible explanations for this are discussed further in section 4.3.

-In the discussion of Fig. 7 in section 4.1, L26 (P9) refers to the consistency between resolutions as "remarkable". It is true that the overall patterns are quite similar, but there are some notable differences in N512, particularly between -15°S–0° off the east coast of Africa. Again, difference fields would be a concise way to highlight similarities and differences.

-From the discussion in section 4.3, it seems the explicit convection experiment is likely getting the right answer for the wrong reasons. The inclusion of this experiment doesn't detract from the main messages of the paper, but I do wonder, what information is to be gained besides motivating future work for repeating this analysis with convective-permitting simulations? Also, consider Molinari and Dudek (1992) "Parameterization of Convective Precipitation in Mesoscale Numerical Models: A Critical Review".

[Figure]

Technical Comments

-P2, L13: Remove "both"

-P2, L14 (and throughout): comma after "e.g."

-P2, L16: Hyphenate "grid-scale" (issue also appears on P13, L14 and L15)

-P3, L10: "MetUM" was defined in the abstract, but has not yet been defined in the main body of text

-P4, L14: "data" should be added between "time-step" and "was"

-P5, L3: Add "(not shown)" between "differences" and "confirming" as this comparison is not included in the paper

-P5, L19: Consider adding "(dashed line)" between "PDFs" and "among" for clarity and reminder for the reader

-P5, L21: Consider adding "strongly" between "not" and "affected" – there are some differences with resolution, albeit not drastic ones

-P6, L8: Consider changing "suggests" to "confirms" as we know N1024 is too coarse for explicit convection

-P6, L19: Is there an extra space before "Switching"?

-P6, L25: It's stated in section 3.1 that differences between resolutions were small, so consider changing the phrasing of this sentence. Maybe "...examine whether the character of grid-box/time-step precipitation discussed in section 3.1 persists..."

-P7, L8: Remove "perhaps" – it is clear that the model show this at a more limited extent than CMORPH

-P10, L29: Add reference to panels "a" and "c" of Fig. 9 to point readers quickly to correct panels

-P13, L24: Add period after "etc"

-Fig. 11, panel (c): should the legend reflect the "N1024e" experiment?

-Overall comment on figures with red and green colored lines: consider changing colors for readers who are red/green colorblind

---

## Author Comment (AC1) · 1 Dec 2016

**GMD-2016-202**

Response to comments by Anonymous Referee 1.

*Specific comments*

- While it is mentioned in your introduction that the representation of rainfall on short timescales is important to longer timescales (because of the biases they incite; Kendon et al. 2014), there is limited discussion of this topic thereafter. I think your hypothesis

attributing the mean-state precipitation biases to issues with intraseasonal variability and sub-daily variability is one of the main takeaways of the paper; therefore, it would be good to include (perhaps in the discussion section) a paragraph highlighting the impact of these biases on the larger scales (e.g., a wet bias in the West Pacific may inhibit MJO propagation).

We agree that this hypothesis is key to our study and was, indeed, the original motivation for the work. While discussion of the impact of mean state biases themselves is outside of the scope of this paper, we have added a few sentences to the Introduction and Discussion sections with additional examples of studies which highlight their impact and the importance of reducing them.

- You noted that many have observed that model biases develop fairly early on in climate simulations. Was this the case with your experiment? Did you observe any sort of spin-up time for the precipitation intermittency? Any note on the lead-dependence of the models' representation of precipitation would be a nice addition.

This is a good suggestion. We find that the intermittency is immediate in our model runs, with no spin-up time at all. We have added this comment to the first point of the Discussion and Conclusions. Further, we note that Klingaman et al. (2016) found similar behaviour in two-day forecasts with an earlier version of the MetUM.

- In section 4.2, you only allude to intraseasonal variability (or the lack thereof) via proxy; that is, we know that if longer and longer temporal averages pull the precipitation spectra to smaller and smaller values, there must be variability on those longer timescales. A power spectral analysis of the observations vs the model, or anything directly showing this intraseasonal variance discrepancy between CMORPH and MetUM, would help drive this point home.

We thank the Reviewer for this comment and we agree that it would be helpful to illustrate this point in a different way. In the new Figure 11, we have used the temporal autocorrelation method from ASoP1 applied to the daily rainfall averaged to the N48

grid over the four regions used in Figure 9 (now Figure 10). This clearly demonstrates the higher day to day persistence of rainfall in the models than the observations over the ocean. The results over the land regions are rather different. In fact, we have corrected the text around discussion of Figure 10d (South China) because the histograms actually suggest too little sub-daily variability (the spectrum of daily averages is similar to that of the 3-hourly averages). The autocorrelations for the "WA" region (new Figure 11b) show noticeable differences between the day to day variability in TRMM and CMORPH, with the day to day variability in TRMM (at this spatial resolution) actually being smaller (higher autocorrelations) than in either the models or CMORPH. Discrepancies in the variability between the two satellite-derived datasets at the 3-hourly timescale were noted in Section 3.2 too (though in the opposite direction). Clearly, the two estimates of actual rainfall amounts have differing characteristics, which must be related both to their different satellite data sources and derivations. It is apparent from the new Figure 11 that, in general, the model tends to over-estimate the persistence of rainfall on this spatial scale in the oceanic regions, but, for the tropical land regions studied, particularly West Africa, we cannot make a definitive statement about the validity of the characteristics of daily rainfall variations in the models compared with satellite-based estimates.

We have added this new Figure and surrounding discussion to Section 4.2.

*Technical corrections*

- P4, L17-20: You use different grammar each time you introduce one of your three subdomains (see the "hereafter" bits)

What we mean here is that data for only those regions were available to us (rather than that we choose which regions to use, as we do for the other simulations). However, we agree that the sentences do not read clearly, so we have changed the sentence to:

For the highest-resolution (N1024) simulations, we use data only over the two limited domains that were available to us (due to storage and computational limits), one in the western Pacific Ocean...

- Include a figure highlighting your subdomains and how they are divided up for section 3.5.

This is now Figure 1.

- P5, L28-32: Add a figure or reference to illustrate/support your explanation of how the all-or-nothing nature of the convective parameterization is the cause of the precipitation intermittency.

We have added this as a "personal communication" from Dr Alison Stirling, who leads our convection parameterization research group. She and her colleagues are currently writing a paper on convective closure which explains this process, but it is not yet ready for submission.

- P6, L30: "of this is what" to "thereof"

Done.

- P7, L2: Did you look at the model output to see if this was the case? Are the differences in precipitation intermittency observable in the raw model output?

Yes we can see this if we plot rainfall time series at a grid-point: time series of time-step data from N1024p have the usual on-off nature, while those from N1024e show continuous (and large) rainfall amounts for 20-30 time steps at a time (1.5-2.5 hours) interspersed by periods of  20 hours of no rainfall. The diagnostic package was designed to characterise and illustrate this behaviour after we (and others) had spent many years looking at time series from individual grid-points!

- P8, L27: "(Fig. 6)" to "(Fig. 6f)"

Done.

- P10, L7-9: These first two sentences of the paragraph should be earlier in the paper, as this is not the first time the data was averaged to N48 and 3-hourly to compare to CMORPH and TRMM (you did this in Figure 5 as well).

Agreed. In fact, we have simply altered these sentences to:

When the precipitation data are all averaged to the N48 grid and 3 h time scale (in a similar manner to Section 3.4), Figure 8c-f shows that the models all tend to underestimate the 3-hourly rainfall amounts compared with TRMM and CMORPH, and that increasing the horizontal resolution does not improve the comparison on this timescale for tropical rainfall over the ocean.

- P10, L24-25: "indicates variability at the longer timescale" is awkward. Perhaps "is due to variability at longer timescales"

We have changed this to "indicates that there is variability at the longer timescale".

*Other changes*

We have updated Figure 2 (new Figure 3) to remove the blank column labelled "XXX" following the reviews of the ASoP1 methods paper by Klingaman et al. (2016). Also following reviews of that paper, and in response to a comment made by Referee 2, we have added a new section 3.6, and associated Table 3, describing summary metrics (from ASoP1) which quantify the differences in spatial and temporal coherence between the datasets.

We have also changed the colours of some of the figures in response to a comment made by Referee 2.

A tracked version of the revised manuscript is included as a supplemental file.

**Supplement:**

[revised manuscript text omitted]

---

## Author Comment (AC2) · 1 Dec 2016

**GMD-2016-202**

Response to comments by Anonymous Referee 2.

*General comments*

This paper presents a useful study on the behavior of tropical precipitation in the MetUM-GA6 model and the sensitivity to grid spacing. The introduction does a nice job setting up motivation for the project and includes a concise synthesis of prior

work related to precipitation modeling. The methods are outlined clearly; however, more justification for the years chosen for analysis could be included. In the results sections, the figures and accompanying text clearly communicate the results; claims are backed up with reasonable explanations and limitations/caveats are noted throughout. The presentation of different types of analyses helps bring together the results of the paper, the main take-away being that in this model, precipitation characteristics are largely unaffected by changing resolutions. Recommendation: Minor Revisions

*Specific comments*

- Discussion of Table 1 on P4 mentions several different geographical domains - a figure outlining all domains used throughout the study would be useful.

This was also suggested by Referee 1 and has been added as a new Figure 1.

- It is mentioned in the Methods discussion that there is little sensitivity in the year chosen for the time-step analysis (P4, L14-15), but is there any justification for why you chose the years you did? For instance, the simulation years for N512 noted in Table 1 are 1982-1990, but 2007 is used for the time-step analysis.

The years were chosen purely according to data availability. Since outputting and storing timestep data is computationally expensive, the diagnostic output is only switched on occasionally and the model year for which this is done is chosen at random. For the N512 simulation, the timestep output was switched on for 1985, but there was a technical problem with the output which meant the time-step diagnostics needed to be switched on again in a repeat of the run. This repeat simulation was started from 2007 due to the availability of a useable 1st June restart file for that year. Comparisons made between different years (on rare occasions when several years of timestep data have been available) have shown little sensitivity to the year chosen.

We have added this information to Section 2.1.

- The discussion of Fig. 1 talks about consistent intermittency between resolutions - this is qualitatively true, but difference PDFs, possibly between the highest and lowest resolutions, or some statistical significance testing could help show this more quantitatively.

We agree with the Reviewer that a more quantitative measure of the differences is necessary. Indeed, following reviews of the paper by Klingaman et al. which describes the ASoP1 methods, summary metrics were added to the methods. Therefore, in the revision of the present manuscript, these have also been used in a new section 3.6 and are shown in Table 3.

- The end of section 3.4 discusses how the explicit convection results compare best to CMORPH/TRMM - consider mentioning that possible explanations for this are discussed further in section 4.3.

Done.

- In the discussion of Fig. 7 in section 4.1, L26 (P9) refers to the consistency between resolutions as "remarkable". It is true that the overall patterns are quite similar, but there are some notable differences in N512, particularly between -15? S-0? off the east coast of Africa. Again, difference fields would be a concise way to highlight similarities and differences.

We did consider this but the problem is that in order to create a difference plot we would have to interpolate the results from the higher resolutions to the lower (or vice versa) and this process would throw away the information that we are trying to present, namely that the spectral shape for the timestep data is almost the same regardless of the resolution. The 1d histograms, which can be overplotted as in Figure 9, are the best way to illustrate this without having to perform any averaging or interpolation.

- From the discussion in section 4.3, it seems the explicit convection experiment is likely

getting the right answer for the wrong reasons. The inclusion of this experiment doesn't detract from the main messages of the paper, but I do wonder, what information is to be gained besides motivating future work for repeating this analysis with convective-permitting simulations? Also, consider Molinari and Dudek (1992) "Parameterization of Convective Precipitation in Mesoscale Numerical Models: A Critical Review".

Indeed, the reason for including this experiment, despite its unrealistic nature, was to demonstrate that the diagnostics were capable of identifying contrasting behaviour between parametrized and explicit convection, in order to motivate their usage for other convection-permitting simulations at more sensible resolutions. We have clarified this point in section 4.3.

*Technical Comments*

- P2, L13: Remove "both"

Corrected.

- P2, L14 (and throughout): comma after "e.g."

Corrected.

- P2, L16: Hyphenate "grid-scale" (issue also appears on P13, L14 and L15)

Corrected.

- P3, L10: "MetUM" was defined in the abstract, but has not yet been defined in the main body of text

Corrected.

- P4, L14: "data" should be added between "time-step" and "was"

Corrected.

- P5, L3: Add "(not shown)" between "differences" and "confirming" as this comparison is not included in the paper

Corrected.

- P5, L19: Consider adding "(dashed line)" between "PDFs" and "among" for clarity and reminder for the reader

Corrected.

- P5, L21: Consider adding "strongly" between "not" and "affected" - there are some differences with resolution, albeit not drastic ones

Corrected.

- P6, L8: Consider changing "suggests" to "confirms" as we know N1024 is too coarse for explicit convection

Corrected.

- P6, L19: Is there an extra space before "Switching"?

Corrected.

- P6, L25: It's stated in section 3.1 that differences between resolutions were small, so consider changing the phrasing of this sentence. Maybe ". . .examine whether the character of grid-box/time-step precipitation discussed in section 3.1 persists. . ."

Corrected.

- P7, L8: Remove "perhaps" - it is clear that the model show this at a more limited extent than CMORPH

Corrected.

- P10, L29: Add reference to panels "a" and "c" of Fig. 9 to point readers quickly to correct panels

Corrected.

- P13, L24: Add period after "etc"

Corrected.

- Fig. 11, panel (c): should the legend reflect the "N1024e" experiment?

Corrected.

- Overall comment on figures with red and green colored lines: consider changing colors for readers who are red/green colorblind

We have now removed the red, pink and yellow colours and replaced with purple and dark brown.

*Other changes*

We have updated Figure 2 (new Figure 3) to remove the blank column labelled "XXX" following the reviews of the ASoP1 methods paper by Klingaman et al. (2016).

A tracked version of the revised manuscript is attached as a supplemental file.